# Beyond Sunk Costs: Boosting LLM Pre-training Efficiency via Orthogonal Growth of Mixture-of-Experts

**Ruizhe Wang** [1 2 †]  **Yucheng Ding** [3 2 †]  **Xiao Liu** [2]  **Yaoxiang Wang** [4 2 †]  **Peng Cheng** [2]  **Baining Guo** [2]
**Zhengjun Zha** [1]  **Yeyun Gong** [2]

## Abstract

As the computational demands for pre-training Large Language Models (LLMs) continue to surge, the need for efficient training paradigms becomes critical. Despite the vast resources already invested in existing pre-trained checkpoints, these assets often remain under-leveraged due to architectural limitations. We introduce an "orthogonal growth" strategy designed to "recycle" these checkpoints by strategically expanding their parameters prior to continued training. Our method focuses on optimizing converged Mixture-of-Experts (MoE) models through two dimensions: interpositional layer copying for increased depth and noisy expert duplication for expanded width. Through extensive scaling laws analysis, we demonstrate a strong positive correlation between the "sunk cost" (prior investment) and the final model accuracy. Empirical results on models up to 70B parameters and 1T tokens show that our recycling approach yields a 10.6% accuracy improvement compared to training from scratch under identical extra compute budgets. This work provides a cost-effective blueprint for sustainable large-scale LLM development.

## 1. Introduction

The unprecedented success of large language models (LLMs) has been largely attributed to scaling laws (Kaplan et al., 2020; Hoffmann et al., 2022), which suggest that increasing model size and training data consistently improves performance. However, training these models from scratch requires enormous computational resources. Consequently, developing methods to scale models efficiently under constrained computational budgets has become a critical research challenge.

Modern LLM development pipelines routinely produce smaller pre-trained model checkpoints and numerous intermediate artifacts from processes like hyperparameter tuning or preliminary evaluations. These models are often discarded once training concludes, leaving much of their potential unrealized due to inherent size constraints. We propose that these checkpoints represent a massive **"sunk cost"**—a significant computational investment that can be systematically leveraged. Model growth offers a new perspective on scaling: rather than starting from scratch, larger models can be created by "recycling" smaller pre-trained models, thereby inheriting their learned knowledge and optimized parameters.

However, recent studies on model growth seldom investigate its application to fully converged models. Existing works (Shen et al., 2022; Du et al., 2024) typically grow models after only a brief initial training period, a scenario that fails to leverage significant sunk costs. This work addresses a more pressing question: what is the optimal method for growing a well-trained model to maximize the return on its substantial sunk cost? Besides, with the increasing adoption of Mixture-of-Experts (MoE) architectures, it is crucial to investigate the effect of model growth on such structures, but to the best of our knowledge, this topic has not been systematically studied until now.

To address this gap, we develop a framework specifically for well-converged MoE models, proposing two orthogonal growth strategies: depth-wise expansion (adding layers) and width-wise expansion (increasing the number of experts), as illustrated in Figure 1 (right). While "stacking" is a common baseline for layer copying (Du et al., 2024; Wu et al., 2024), we provide empirical evidence that our "interpositional" method is superior. Our analysis reveals a critical threshold: models trained with more than Chinchilla-scaling law's (Hoffmann et al., 2022) optimal tokens exhibit unique inter-layer weight norm patterns. We show that while stacking disrupts these established structures, the interpositional

---

[†]Work done during internship in MSRA [1]University of Science and Technology of China [2]Microsoft Research Asia [3]Shanghai Jiao Tong University [4]Xiamen University. Correspondence to: Yeyun Gong <yegong@microsoft.com>.

*Proceedings of the 43rd International Conference on Machine Learning*, Seoul, South Korea. PMLR 306, 2026. Copyright 2026 by the author(s).

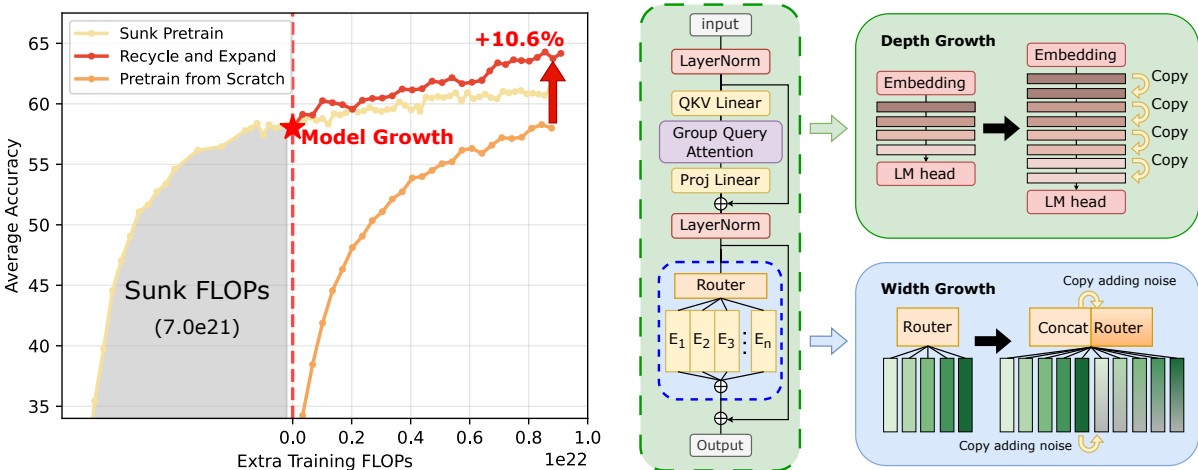

*Figure 1.* An overview of the orthogonal growth framework for Mixture-of-Experts (MoE) models. The left panel shows the training efficiency gain compared to training from scratch. The right panel illustrates the two orthogonal growth dimensions: Depth Growth via layer copying and Width Growth via expert expansion with noise.

method maintains them, thereby achieving better performance in deeply converged models. Moreover, we discover that adding a small amount of noise to newly copied experts is crucial, as it facilitates better expert specialization. We also empirically validate the orthogonal nature of the two proposed growth strategies, ensuring consistent performance regardless of their execution sequence.

We also provide a comprehensive study on the optimal timing for growth to best utilize the sunk cost. Our findings reveal a strong positive correlation between the amount of sunk FLOPs and the final performance of the grown model. This confirms that a greater initial investment leads to a better final model, highlighting the efficacy of our framework in recycling prior computation. We further demonstrate that under a fixed total training budget (sunk + additional FLOPs), model growth is comparable or even slightly superior to training a large model from scratch.

Finally, we conduct extensive experiments to demonstrate the scalability and robustness of our orthogonal growth framework. As shown in Figure 1 (left), our method effectively scales an MoE model from 17 billion to 70 billion parameters using a 1-trillion-token dataset. The resulting model achieves a 10.6% average accuracy improvement on downstream tasks compared to a model trained from scratch with the same additional FLOPs budget.

## 2. Related Work

**Efficient Pretraining**. Cost reduction in pretraining follows two paths: direct computational savings via quantization (Jacob et al., 2018; Micikevicius et al., 2017; Peng et al., 2023; Wang et al., 2025), pruning (Zhu & Gupta, 2017;

Xia et al., 2022; Ma et al., 2023), and distillation (Gou et al., 2021; Loureiro et al., 2021; Sreenivas et al., 2024); or reusing sunk costs through model growth (Shen et al., 2022) and upcycling (Komatsuzaki et al., 2023; Liew et al., 2025).

**Model Growth for Pretraining**. Model expansion increases parameters during or after training. Early works focused on CNNs (Chen et al., 2015), BERT (Chen et al., 2022; Gong et al., 2019; Yao et al., 2024), or ViTs (Wang et al., 2024b; 2023). For Transformers, though growth and initialization are explored (Shen et al., 2022; Du et al., 2024; Wang et al., 2024b), studies often use small-scale data. In the LLM era, LLaMA Pro (Wu et al., 2024), Solar 10.7B (Kim et al., 2024), and FLM-101B (Li et al., 2023) applied expansion to larger scales, but with limited technical analysis. Theoretical understanding of depth expansion has progressed through the connection between residual networks and neural ODEs (Marion et al., 2024), and through practical scaling of deep Transformers (Wang et al., 2024a). We extend this paradigm to MoE architectures and well-converged models (discussed in Section 3.1).

**MoE Model Upcycling**. MoE (Shazeer et al., 2017; Zhou et al., 2022; Mu & Lin, 2025) scales capacity with sparse computation. "Upcycling" initializes MoE from dense checkpoints (Komatsuzaki et al., 2023; Team, 2024; Wei et al., 2024), often involving expert granularity adjustments (Nakamura et al., 2025; He et al., 2024). Unlike upcycling, which requires random router initialization and high noise (50%+) for expert divergence (Komatsuzaki et al., 2023; Muennighoff et al., 2024; Nakamura et al., 2025), our method expands existing MoE models. By reusing pretrained routers, we require only minimal noise, as excessive noise becomes counterproductive (discussed in Section 3.2).

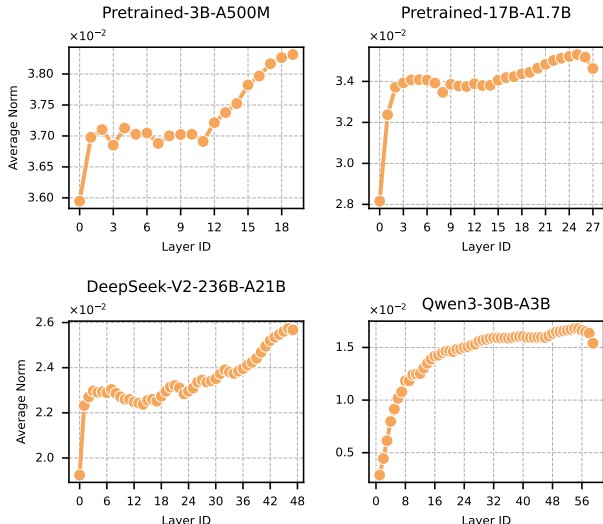

*Figure 2.* Characteristic layer-wise weight norm distribution in pre-trained LLMs, including pre-trained models in this work and from open-source community.

## 3. Methodology

This section introduces orthogonal growth strategies for Mixture-of-Experts models. Section 3.1 introduces **Depth Growth**, a method for expanding a model by duplicating its layers. Section 3.2 presents **Width Growth**, which involves expanding the number of experts. Section 3.3 compares these two strategies and investigates the orthogonality.

### 3.1. Depth Growth

Large Language Models (LLMs) are typically constructed from multiple transformer layers. Given a model $m$ with layers $l_1, l_2, \ldots, l_n$, the widely used **stacking** method (Wu et al., 2024; Du et al., 2024) concatenates the original layers sequentially $k$ times:

$$M = \text{stack}(m)$$
$$= \underbrace{l_1, l_2, \ldots, l_n, \ l_1, l_2, \ldots, l_n, \ \cdots, \ l_1, l_2, \ldots, l_n}_{k \text{ times}} \quad (1)$$

Alternatively, the **interposition** method duplicates each layer k times in place:

$$M = \text{interposition}(m)$$
$$= \underbrace{l_1, l_1, \ldots, l_1}_{k \text{ times}}, \ \underbrace{l_2, l_2, \ldots, l_2}_{k \text{ times}}, \ \cdots, \ \underbrace{l_n, l_n, \ldots, l_n}_{k \text{ times}} \quad (2)$$

While prior work favors stacking, it primarily examines early-stage training where layer distributions remain similar. We hypothesize that for well-converged checkpoints,

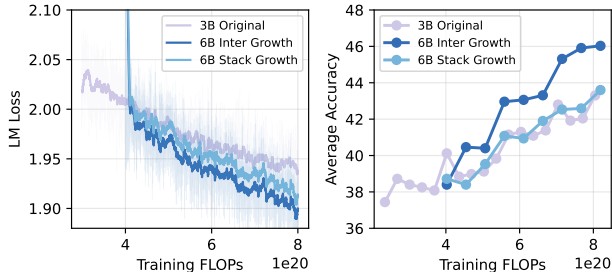

*Figure 3.* Performance comparison of interposition and stack depth growth strategies. Left: training loss; Right: average downstream task accuracy.

stacking **disrupts** learned functional structures, whereas interposition **preserves** them.

**Structural Observations**. As shown in Figure 2, pre-trained LLMs exhibit a characteristic layer-wise weight norm[1] pattern: small and variable in the initial layers, monotonically increasing through the middle, and slightly decreasing near the final layers. This trend is consistently observable across our models and several popular open-source models (additional examples in Appendix A). Recent theoretical work offers a principled explanation: Marion et al. (2024) proved that gradient-flow-trained ResNets are implicitly regularized toward neural ODEs, implying that the per-layer transformation $x_{l+1} = x_l + f_l(x_l)$ should form a *smooth* sequence; Wang et al. (2024a) further showed that stable deep training requires controlling the growth of residual contributions across layers. Together, these results suggest the observed norm profile is a structural signature of healthy convergence. When grown from such converged checkpoints, we should therefore strive to maintain this monotonic trend. The stacking method disrupts this position-dependent structure by concatenating early-layer norms after late-layer norms, whereas interposition preserves it by duplicating each layer in place. We further show below that this norm pattern is directly tied to the model's convergence status.

**Comparative Experiments**. We evaluated these strategies by growing a 3B MoE model (700M active parameters) to 6B (1B active), with the growth factor $k$ in Equation (2) fixed to 2 following peer work (Shen et al., 2022; Wang et al., 2024b).[2] Details on model structure and training settings are available in Appendix F. Results in Figure 3 demonstrate that interposition consistently yields lower training loss and higher downstream accuracy[3]. To ensure a fair com-

---

[1]Average Frobenius norm (i.e. L2 norm of flattened 2D tensor) of expert weight matrices, normalized by square root of number of elements.

[2]Discussion on growth factor can be found in Appendix B.

[3]The accuracy metric is computed as the average score across multiple downstream evaluation tasks, such as MMLU, ARC-C, HellaSwag, BoolQ and OpenbookQA. The computation method is detailed in Appendix G.1, and full results are in Appendix G.2.

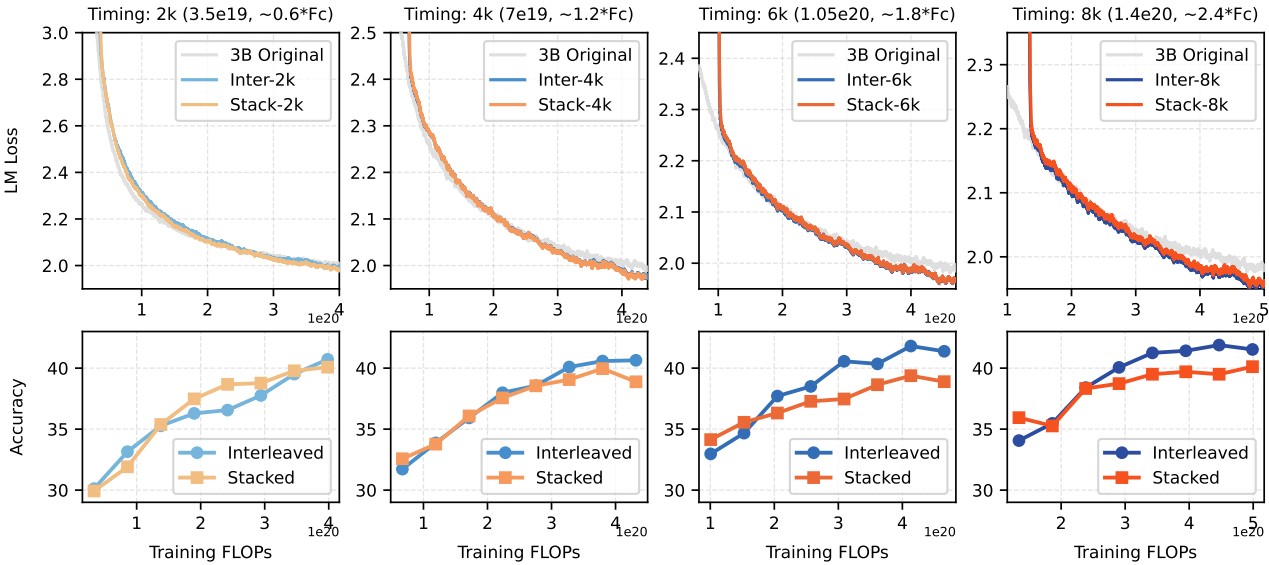

*Figure 4.* Investigation on the boundary condition for interposition method's superiority over stack method, provided with loss curves and accuracy evaluation results based on different growth timing.

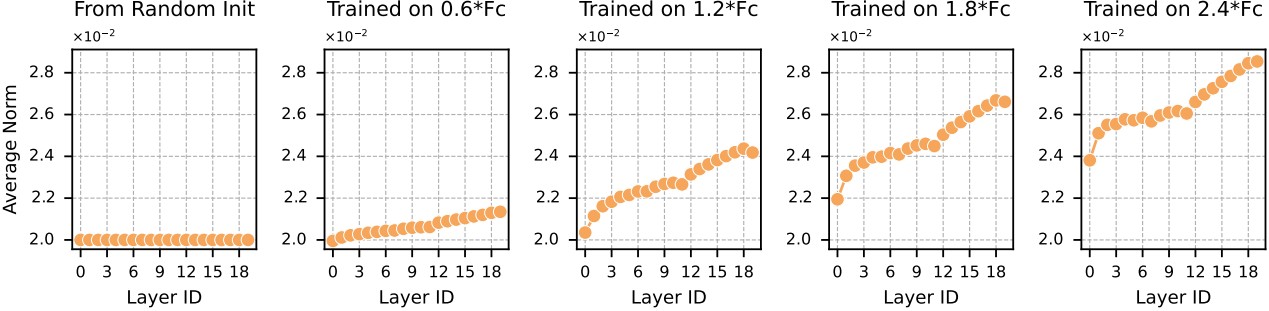

*Figure 5.* Investigation on the variation trend for average weight norm during model training process.

parison given the increased size of the grown models, the x-axis represents the total training Floating Point Operations (FLOPs).

**The Convergence Boundary**. To quantify the boundary condition for the superiority of interposition method, we conducted a systematic study using the **amount of training FLOPs** as a measure of training progress. Chinchilla Scaling Law (Hoffmann et al., 2022) gives a compute-optimal ratio between model size (N) and token count (D): $D = 20N$. The corresponding total compute can be estimated as $6ND$ where the factor 6 follows from the empirical rule of forward + backward computation. For MoE models, the situation is more complicated, but a reliable approximation is to base the compute on the number of activated parameters instead of the total parameter count (Fedus et al., 2022). For our 3B model ($N_a = 700M$), the resulting compute-optimal FLOPs ($F_c$) is approximately $F_c \approx 6 \cdot N_a \cdot (20N_a) = 5.88 \times 10^{19}$. In practice, small models are commonly overtrained well beyond this compute-optimal value.

Using the accumulated FLOPs of the 3B-A700M model as the indicator, we examined checkpoints at various training stages. At each selected checkpoint, we applied either stack or interpositional layer growth, then continued training for a fixed number of steps and compared their performance. Based on the findings in Figure 4, we observe that the critical point at which the two growth methods begin to diverge emerges at approximately **1× the Chinchilla-optimal FLOPs** ($F_c$). This suggests that once the total training FLOPs exceed $F_c$, the interpositional method should be preferred over the stack method, indicating that the model has reached a well-converged state.

**Correlation with Weight Norms**. To further connect this observation with our findings on weight norms, Figure 5 shows the weight norm distributions of the checkpoints used in this experiment. In the beginning, the weights are initialized from an i.i.d. Gaussian distribution with a fixed standard deviation ($\delta = 0.02$), leading to uniform layer-wise norms. As training progresses, the weight-norm distribution develops a clear upward trend. Around 4k steps ($\approx 1.2F_c$),

*Table 1.* Per-layer average expert weight norm after growth. Stacking introduces a sharp norm discontinuity at L19→20 (∼10× average inter-layer gap), while interposition preserves the smooth profile.

| | | **Stack** | | **Interposition** | |
|---|---|---|---|---|---|
| **Layer** | Base | @0 | +16k | @0 | +16k |
| 0 | .0323 | .0323 | .0352 | .0323 | .0349 |
| 10 | .0336 | .0336 | .0363 | .0337 | .0363 |
| **19** | **.0356** | **.0356** | **.0373** | .0336 | .0362 |
| **20** | — | **.0323↓** | **.0367↓** | .0336 | .0362 |
| 30 | — | .0336 | .0373 | .0347 | .0370 |
| 39 | — | .0356 | .0373 | .0356 | .0376 |

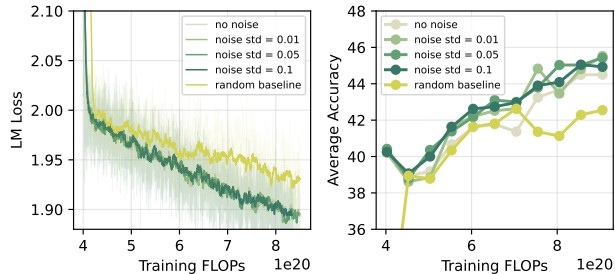

*Figure 6.* The impact of noise injection scale on width growth performance. Left: training loss; Right: average downstream task accuracy.

a stable layer-wise increasing pattern emerges, and in subsequent training the norms continue to grow while maintaining this structure. Therefore, we believe that **the emergence of this characteristic increasing pattern across layers marks the boundary at which interpositional growth begins to outperform stack growth**. In conclusion, for converged models rather than those in the early stages of training, the interpositional method is a better choice than the widely adopted stacking method.

Further, for our converged 3B model, the norm drop from layer 19 to the stacked layer 20 is approximately 10× the average inter-layer variation (Table 1). The model must then expend training budget repairing this discontinuity. In contrast, interposition doubles the trajectory's resolution while maintaining continuity, as each duplicated layer pair preserves the local norm structure. This empirical evidence further corroborates our earlier hypothesis that the layer-wise weight norm profile is predictive of growth strategy effectiveness.

### 3.2. Width Growth

For MoE models, an alternative to increasing depth is to expand the parameter count by increasing the number of experts. To preserve the capabilities of a converged MoE model during such growth, it is crucial to proportionally increase the number of activated experts (the top-k parameter) as tokens must be routed to the newly added capacity.

In our width growth experiments, we simultaneously double the total number of experts ($E \to 2E$) and the activated experts ($k \to 2k$).[4] For an original MoE layer $F$ with a top-$k$ routing scheme, the output is defined as:

$$F(x) = \sum_{i \in \mathcal{T}} g_i(x) f_i(x), \quad \mathcal{T} = \text{Top}_k(g(x)) \quad (3)$$

where $f_i$ denotes the $i$-th expert and $g(x) \in \mathbb{R}^E$ represents the gating weights. To expand this to $F_g(x)$ with $2E$

experts and top-$2k$ routing, we first replicate the existing experts to inherit the model's learned capabilities. To facilitate expert divergence and specialization while maintaining structural stability, we introduce a perturbation-based initialization. Specifically, we initialize the new experts and their corresponding router weights by adding Gaussian noise $\epsilon \sim \mathcal{N}(0, (\alpha\sigma_{\text{orig}})^2)$, where $\sigma_{\text{orig}}$ is the standard deviation of the original parameters. The expanded weights are then formed by concatenating the original and perturbed weights. We empirically set a minimal noise scale (e.g., $\alpha = 0.01$) to encourage functional differentiation without destabilizing the well-converged original experts.

As illustrated in Figure 6, while direct expert replication ($\alpha = 0$) and our noise-augmented method yield comparable language modeling loss, the latter achieves better performance on downstream tasks, with an accuracy gain of approximately 1%. The results also indicate that excessive noise may be harmful. Furthermore, a baseline on randomly initializing the new experts while retaining the original ones, yields significantly poorer performance, validating that the observed benefits stem from knowledge inheritance via noisy copies rather than simple symmetry breaking. These findings underscore the importance of adding a small magnitude of noise to stimulate functional specialization among experts during width growth. Intuitively, exact duplication creates perfect symmetry: all copied experts receive identical gradients from the router, preventing functional divergence. A small perturbation breaks this symmetry while preserving the learned representations, allowing the router to gradually differentiate experts during continued training. This mechanism is fundamentally different from the large-noise regime used in MoE upcycling from dense models (Komatsuzaki et al., 2023; Nakamura et al., 2025), where randomly initialized routers require substantial perturbation ($\geq$50%) to force expert divergence.

### 3.3. Discussion on Depth and Width Growth

Depth and width expansions represent two orthogonal dimensions for scaling MoE models. As illustrated in Figure 7,

---

[4]Discussion on routing variants can be found in Appendix C.

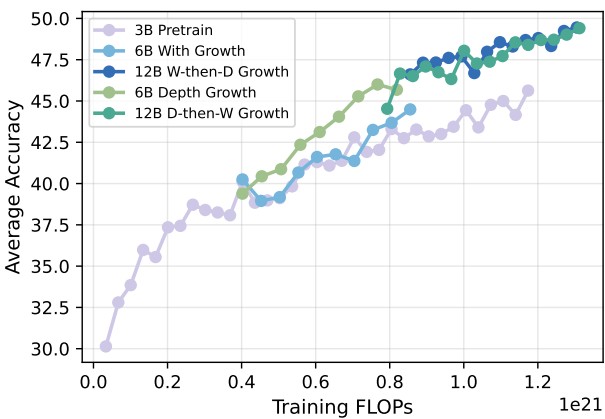

*Figure 7.* Performance comparison of depth growth, width growth and with two different orders of growth.

for single-axis growth, depth growth generally yields better downstream performance compared to width growth. We attribute this to the fact that newly added experts in width growth require more extensive training to achieve effective load balancing and functional specialization, making its performance gains less immediate. Conversely, width growth is more effective at preserving model stability, as it aligns more closely with the principles of **Function-Preserving Transformations** (Evci et al., 2022; Wang et al., 2023; Yao et al., 2024). We provide a more detailed analysis of this structural stability in Appendix D.

Furthermore, Figure 7 demonstrates the robustness of dual-operation growth. Specifically, the final performance is invariant to the sequence of expansion (i.e., width-then-depth vs. depth-then-width) as both paths converge to equivalent performance levels. This observation reinforces our hypothesis that depth and width growth are mutually orthogonal and complementary, allowing for flexible integration in large-scale pre-training, as further demonstrated by our scaling experiments in Section 5.

**Empirical Orthogonality Analysis.** Beyond order-independence, we verify that the two growth paths are orthogonal in optimization dynamics. On shared parameters, Adam's first-moment cosine similarity between the pre-growth model and each grown variant remains near zero throughout training (all $|\cos| < 0.04$, Table 2), confirming that each growth operation redirects optimization into a largely independent subspace. Furthermore, the cosine similarity of cumulative weight updates between depth-grown and width-grown models decreases monotonically (Table 3), with MoE-specific parameters exhibiting the strongest divergence. Full per-layer analysis is in Appendix E.

## 4. Analysis of Growth Timing and Sunk Cost

Having established the efficacy of our growth methods, we now turn to a critical practical question: when is the optimal

*Table 2.* Adam first-moment cosine similarity between pre-growth and post-growth models on shared parameters.

| **Steps** | $\cos(\mathbf{m}_{\text{base}}, \mathbf{m}_{\text{depth}})$ | $\cos(\mathbf{m}_{\text{base}}, \mathbf{m}_{\text{width}})$ |
|---|---|---|
| +2k | 0.031 | −0.003 |
| +4k | 0.039 | 0.022 |
| +6k | 0.012 | 0.034 |
| +8k | −0.001 | 0.027 |
| +10k | 0.012 | 0.001 |
| +12k | −0.009 | −0.018 |
| +14k | −0.008 | 0.011 |
| +16k | 0.016 | 0.038 |

*Table 3.* Cosine similarity of cumulative weight updates ($\Delta W$) between depth-grown and width-grown models on shared parameters.

| **Steps** | **Overall** | **Expert FFN** | **Attention** | **Embedding** |
|---|---|---|---|---|
| +2k | 0.630 | 0.614 | 0.635 | 0.781 |
| +8k | 0.490 | 0.469 | 0.562 | 0.725 |
| +16k | 0.431 | 0.406 | 0.551 | 0.713 |

time to apply them? In this section, we investigate the optimal point during the pre-training process to apply the growth strategy and compare its efficacy against training a larger model from scratch. We demonstrate that even for already converged trained checkpoints, model growth can still effectively leverage the computational investment (i.e. sunk FLOPs cost).

### 4.1. Impact of Sunk Cost with a Fixed Additional Budget

This analysis addresses a primary question: given a series of checkpoints with varying amounts of sunk cost, which serves as the optimal base for growth? Specifically, does a greater sunk cost lead to superior performance post-growth?

Based upon the methodology in Section 3, we pretrain the 3B MoE model to full convergence using a standard schedule encompassing warmup, constant, and annealing phases (Figure 9). To systematically evaluate the utility of sunk costs, we extracted a series of checkpoints representing diverse training stages and subjected each to a fixed budget of additional training FLOPs. Given that depth expansion consistently outperforms width expansion, these scalability experiments focus exclusively on depth growth. Specifically, we selected 12 checkpoints sampled between 8k and 96k steps, growing each to 6B parameters. We also include a 6B model trained from scratch as a baseline, effectively representing a growth scenario with zero sunk FLOPs.

The results of growing models from different checkpoints, each with the same budget for additional FLOPs, are shown in Figure 8. Both the final training loss and the average

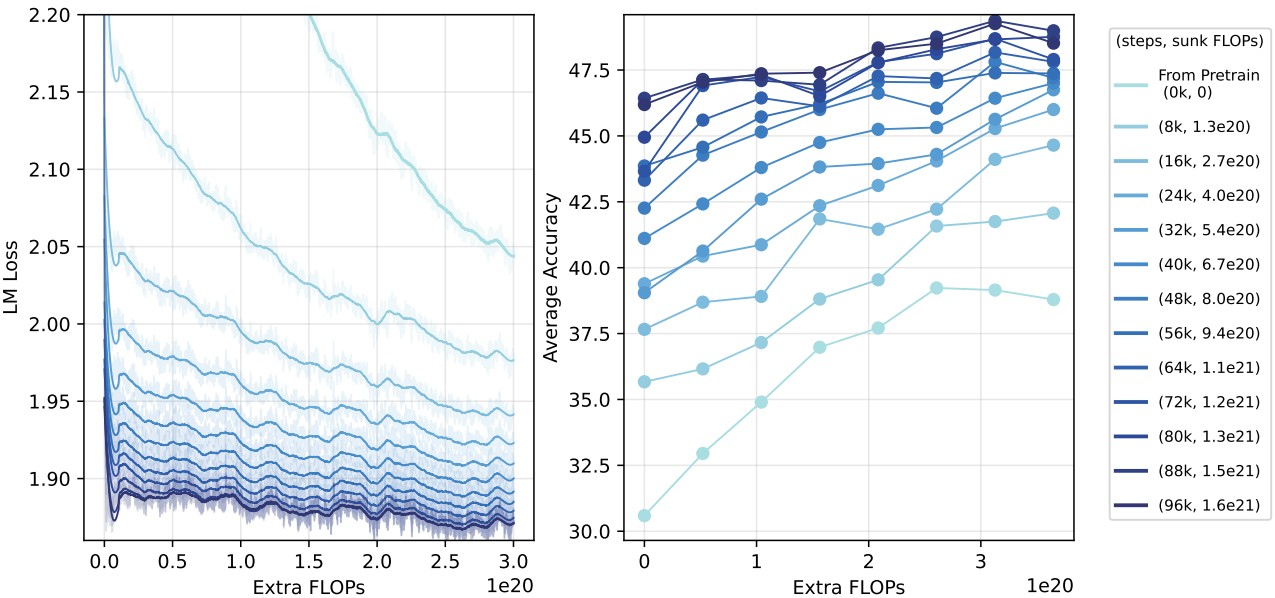

*Figure 8.* Investigation of growth time according to **amount of sunk cost**. Left: loss curve. Right: average downstream task accuracy.

*Table 4.* Quantitative accuracy results growth time investigation for amount of sunk cost.

| Start steps | 0k | 8k | 16k | 24k | 32k | 40k | 48k | 56k | 64k | 72k | 80k | 88k | 96k |
|---|---|---|---|---|---|---|---|---|---|---|---|---|---|
| Start acc | 30.59 | 35.67 | 37.66 | 39.39 | 39.05 | 41.11 | 42.26 | 43.86 | 43.32 | 43.66 | 44.95 | 46.43 | 46.19 |
| End acc | 38.79 | 42.07 | 44.65 | 46.00 | 46.75 | 47.00 | 47.20 | 47.37 | 47.81 | 47.90 | 48.76 | 48.99 | 48.52 |
| **Average acc** | 36.29 | 39.09 | 41.19 | 42.69 | 43.34 | 44.51 | 45.67 | 46.15 | 46.49 | 47.13 | 47.43 | 47.88 | 47.82 |

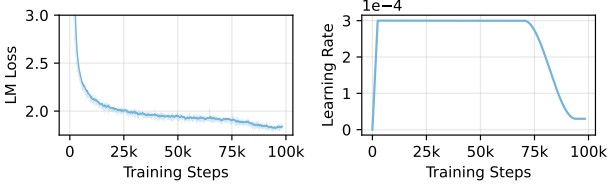

*Figure 9.* Full training curve and learning rate scheduler of 3B model pretraining.

downstream accuracy exhibit a **strong positive correlation** with the sunk cost invested prior to growth. This indicates that a larger initial training investment leads to a better final model, confirming that the growth method effectively recycles prior computational work, and suggests that later checkpoints with more sunk cost can be leveraged to achieve better growth performance. We further present quantitative results in Table 4 to support this positive correlation. The table reports the starting, ending, and average accuracy across the entire continued training process with an additional $3 \times 10^{20}$ FLOPs.

Notably, while the positive correlation persists when the base model enters the learning rate annealing stage (beyond

72k steps), the marginal performance gains diminish. This is likely because all grown models in this experiment are trained with the same new constant learning rate ($3 \times 10^{-4}$) for fair comparison, which may not be optimal for a checkpoint from a late annealing phase. This suggests that *one should either carefully tune the learning rate for the continued training phase or, preferably, select a checkpoint from the constant learning rate phase for growth.*

### 4.2. Comparison to Scratch Training with a Fixed Total Budget

The results in Figure 8 also demonstrate that, for a fixed *additional* training budget, model growth is clearly superior to training from scratch. We next investigate whether this advantage holds when the *total* FLOPs budget is fixed. The results of this experiment are presented in Figure 10. Here, models grown from later checkpoints are allocated a correspondingly smaller budget for continued training.

The results show that for most growth timings, the final accuracy of the grown model is comparable or slightly superior to the scratch-trained model. Specifically, models grown from earlier checkpoints, which thus allocated a larger proportion

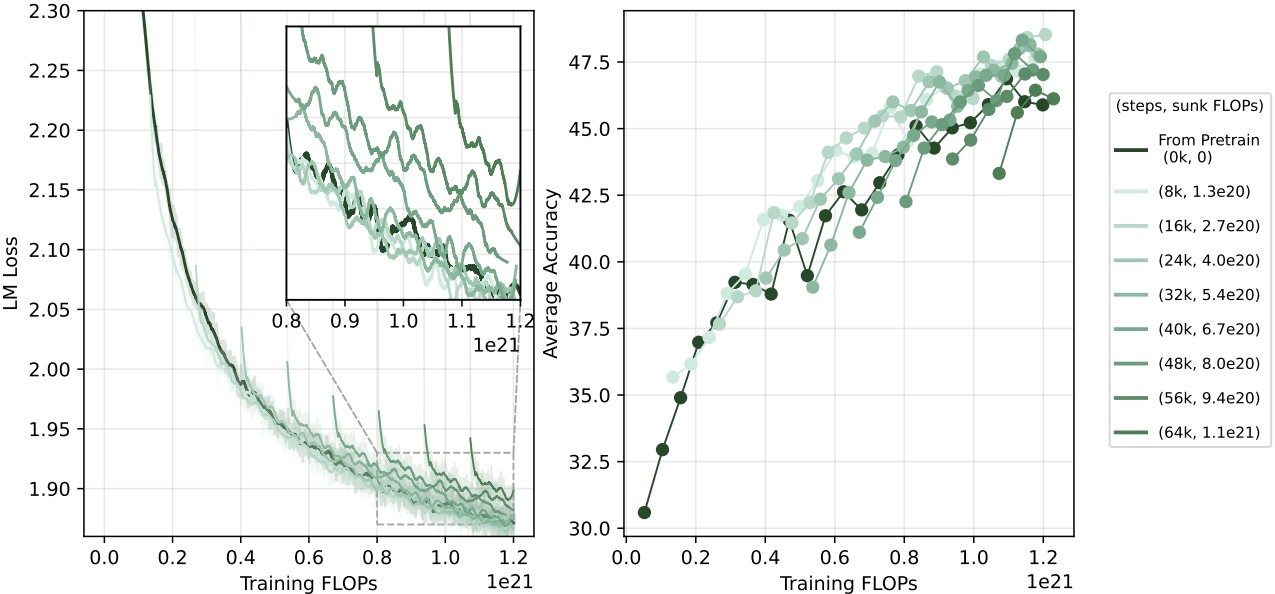

*Figure 10.* Investigation of growth time according to **total amount of cost**. Left: loss curve. Right: average downstream task accuracy.

*Table 5.* Quantitative accuracy results growth time investigation for total amount of cost.

| Start steps | 0k | 8k | 16k | 24k | 32k | 40k | 48k | 56k | 64k |
|---|---|---|---|---|---|---|---|---|---|
| End acc | 45.03 | 47.66 | 48.53 | 47.80 | 48.15 | 47.70 | 47.20 | 47.03 | 46.12 |
| **Average acc** | 45.82 | 46.99 | **47.38** | 47.29 | 47.05 | 46.96 | 46.47 | 45.74 | 45.37 |

of the total budget for post-growth training, tend to perform best. This suggests that the pre-trained smaller model serves as a highly effective initialization for the larger model's training process. The growth method underperforms only when initiated from a very late checkpoint, where the budget for continued training is insufficient. This provides a valuable heuristic: *one should allocate additional FLOPs at least on the same order of magnitude as the sunk cost in order to achieve performance comparable to pre-training under the same total FLOPs*.

Quantitative results are provided in Table 5 to support this finding, showing the average accuracy over the final six accuracy measurements (with the exception of line 64k, which contains only four data points). Notably, although the training loss of later checkpoints remains relatively high, the final accuracy quickly recovers during continued training.

In conclusion, model growth is an effective strategy for leveraging the sunk cost of pre-trained models, with final performance positively correlating with the initial training investment. Furthermore, its effectiveness is comparable and sometimes superior to training from scratch, even when evaluated under a fixed total-FLOPs budget.

## 5. Scalability Experiments

The practical utility of model growth depends on its scalability, particularly as training larger models entails substantially higher sunk costs. To evaluate this, we scale our framework to a 17B-parameter MoE model, which is progressively expanded to 70B parameters over one trillion training tokens. This large-scale validation underscores the robustness and effectiveness of our growth paradigm in real-world, high-compute scenarios.

Following the methodologies in Section 3, we first re-examine our findings on depth growth at the 17B scale. The 17B base model (2.1B activated parameters) serves as a scaled-up version of the 3B architecture; comprehensive architectural and training specifications are detailed in Appendix F. We pre-train this model with $7 \times 10^{21}$ FLOPs (approximately $13 \times F_c$, $F_c \approx 5.3 \times 10^{20}$), ensuring a state of deep convergence. We then compare the "stacking" and "interpositional" methods. As illustrated in Figure 12, the interpositional method consistently outperforms stacking, further substantiating our insight that preserving learned structural patterns is critical when expanding well-converged models.

In this scalability study, we sequentially expand the model's

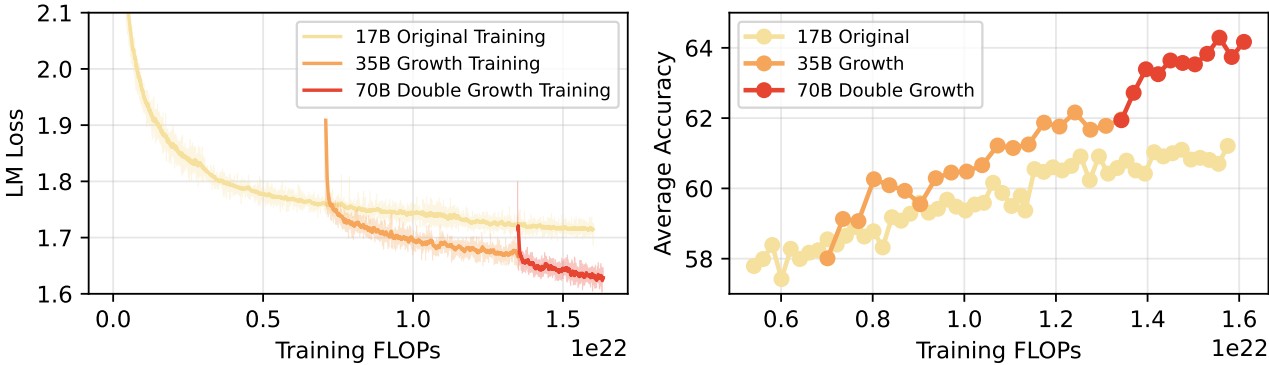

*Figure 11.* Full training loss and downstream task evaluation result for 17B model pretraining and growth training. Left: training loss; Right: average downstream task accuracy.

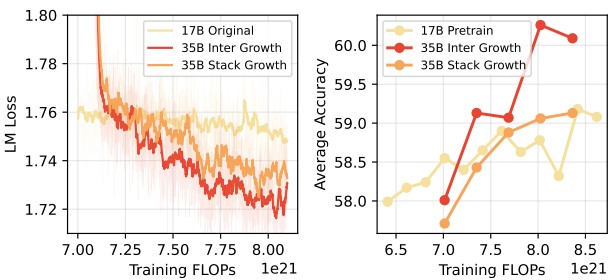

*Figure 12.* Performance comparison of interposition and stack depth growth strategies for 17B model. Left: training loss; Right: average downstream task accuracy.

depth and width to enhance its functional capacity and expert specialization, respectively. Leveraging the orthogonality demonstrated in Section 3.3, this tiered growth strategy allows us to achieve significant scaling without the prohibitive cost of redundant experimentation. The process begins with the initial 17B model (6 / 96 experts activated) trained for approximately 600B tokens. We first apply Depth Growth, doubling the layers from 28 to 56 to produce a 35B intermediate model. Following an additional 300B tokens of training, we execute Width Growth, doubling the expert count from 96 to 192. This results in the final 70B configuration, which is trained for a final 100B tokens (achieving total 1T tokens). The comprehensive training loss trajectory and downstream performance metrics are detailed in Figure 11.

Our experimental results reveal a critical insight: model growth unlocks substantial performance gains even after the base model's improvement has saturated from extensive pre-training. As shown in Figure 11, the final 70B model achieves an average accuracy of 64.17, marking an improvement of 2.21 points over the 35B checkpoint and 5.62 points over the initial 17B model. When controlled for total training FLOPs, the 70B model still outperforms the 17B baseline by 2.96 points (a 4.0% relative gain). From the perspective of sunk cost utilization, our growth strategy demonstrates superior efficiency, surpassing the from-

scratch baseline by 6.18 points (a 10.6% relative gain) under the same extra FLOPs budget, as illustrated in Figure 1. These large-scale results reaffirm that model growth is a powerful strategy for leveraging existing computational investments while effectively pushing the performance boundaries of converged LLMs.

# 6. Conclusion

This work presents a systematic framework for model growth, addressing the computational cost problem in Large Language Model (LLM) pre-training. We demonstrate that pre-trained checkpoints, often considered disposable intermediate assets, can be effectively "recycled" to create larger and more capable models, thus preserving their significant sunk cost. Our research identifies optimal strategies for two orthogonal growth dimensions in Mixture-of-Experts (MoE) models, establishes a scaling principle that growing from a more converged checkpoint yields superior final performance, and shows that our framework is highly scalable. These methods contribute to a more computationally sustainable ecosystem, paving the way for more efficient and democratized access to frontier-scale model pre-training.

## Impact Statement

This research focuses on developing methods for efficiently scaling large language models through model growth, with the primary goal of reducing computational costs and reusing previously trained checkpoints. The study does not involve human subjects, personal data, or sensitive demographic information. Potential societal impacts of large language models are acknowledged, such as misuse for generating harmful or biased content, but this work does not introduce new risks beyond those already inherent in the use of such models. Instead, by improving training efficiency, the proposed methods may lower the environmental footprint of model development.

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

## A. More Results on Layer-wise Norm Distribution

We further extend our analysis on the layer-wise weight norm distribution trend by examining a broader range of open-source MoE models. Specifically, we compute and visualize the layer-wise average weight norm distributions for Deepseek-v2-Lite-16B-A2.4B (DeepSeek-AI, 2024), Qwen1.5-MoE-14.3B-A2.7B-Chat (Yang et al., 2025), Mixtral-8x7B (Jiang et al., 2024), Hunyuan-A13B-Instruct (Sun et al., 2024), Dots-LLM1-142B-A14B (Huo et al., 2025), and GroveMoE-Inst-33B-A3.2B (Wu et al., 2025).

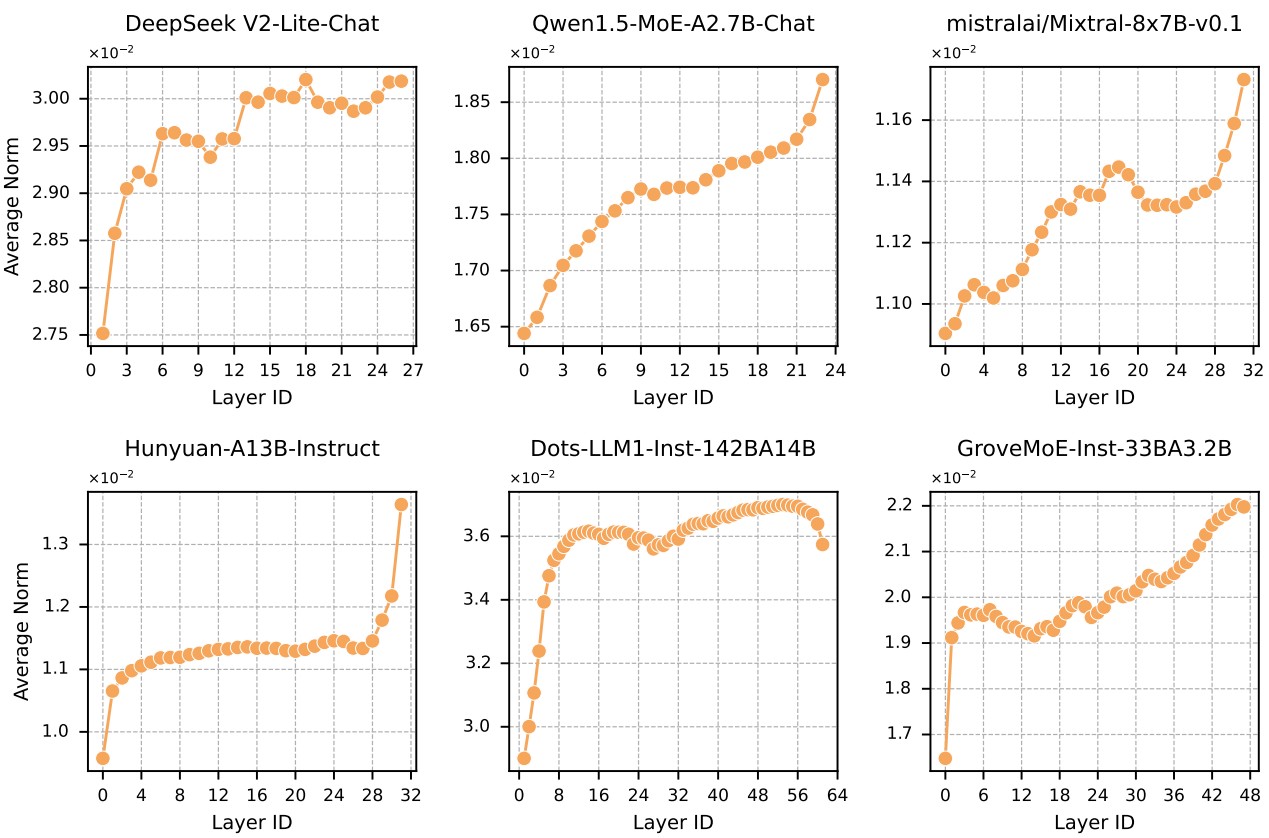

*Figure 13.* Characteristic layer-wise weight norm distribution in pre-trained LLMs from several open-source models.

As shown in Figure 13, a consistent pattern emerges across well-converged MoE models: the layer-wise weight norms tend to increase with depth. This trend provides further empirical support for our proposed interpositional growth method (Section 3.1), highlighting its ability to align with the intrinsic training dynamics of large MoE architectures.

## B. Discussion on Effect of Growth Factor $k$ on Depth Expansion

The advantage of the interposition method is rooted in its ability to preserve learned weight-norm trajectories during model expansion. While the primary experiments utilize a growth factor of $k = 2$ following prior literature, we further investigate the robustness of this strategy under larger expansion factors. Theoretically, larger $k$ values exacerbate the structural disruptions caused by the stacking method, often leading to erratic "rise-and-fall" weight-norm patterns. In contrast, the interposition method maintains a stable, non-decreasing norm trend even as model depth increases significantly.

To empirically evaluate the impact of the growth factor, we conducted additional experiments by setting $k = 4$, expanding the 20-layer 3B MoE model into an 80-layer architecture. The training trajectories and downstream performance are illustrated in Figure 14.

The results demonstrate that the superiority of interposition over stacking persists at $k = 4$. The performance gap reinforces the hypothesis that preserving the pre-trained weight-norm trajectory is critical for efficient growth, particularly in extremely

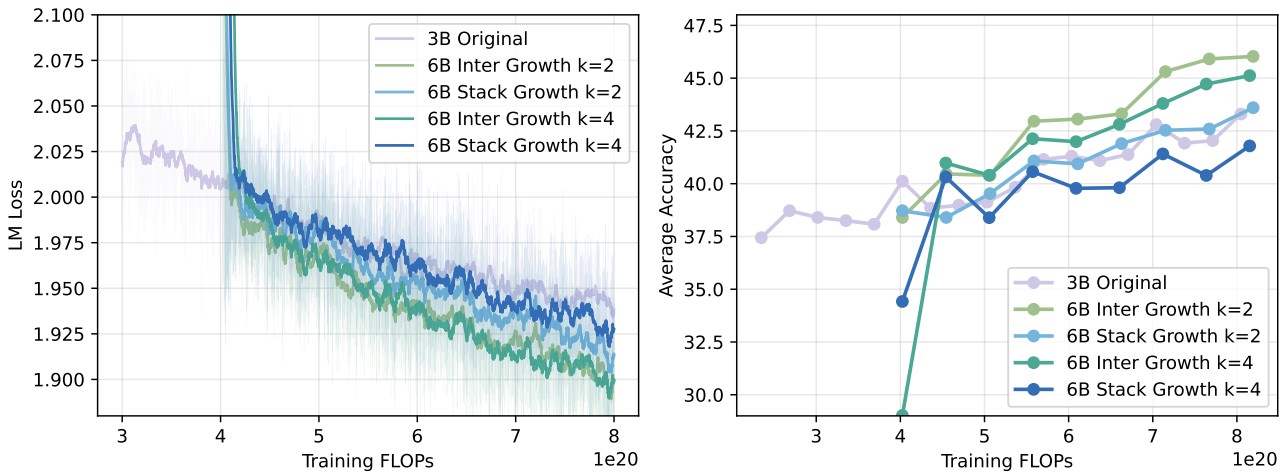

*Figure 14.* Performance comparison of interposition and stacking strategies across different growth factors ($k = 2$ and $k = 4$). Left: training loss; Right: average downstream task accuracy.

deep configurations where stacking-induced disruptions become more pronounced.

## C. Discussion on Routing Variants in Width Growth

To justify our strategy of simultaneously doubling the total number of experts ($E$) and the number of activated experts ($k$), we evaluate several alternative routing configurations. Building upon our baseline 3B-A700M (top-4/64) and the width-grown 6B-A900M (top-8/128), we introduce two additional variants: 6B-A700M (top-4/128) and 3B-A900M (top-8/64).

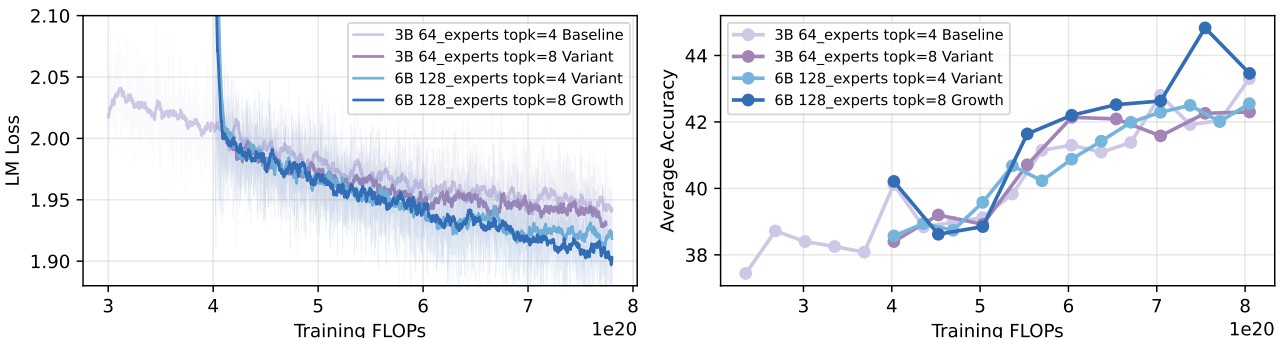

*Figure 15.* Performance comparison of top-$k$ variants during width growth. Left: training loss; Right: average downstream task accuracy.

6B-A700M (Expansion without increased compute): In this variant, we expand the total expert count to 128 while maintaining $k = 4$. This model underperforms compared to the 6B-A900M baseline, suggesting that architectural expansion without a corresponding increase in active computation fails to yield effective gains in training efficiency.

3B-A900M (Increased compute without expansion): Here, we double $k$ from 4 to 8 without adding new parameters. While this variant initially shows reasonable gains, its subsequent convergence rate is significantly slower. This indicates that increasing computation within a fixed capacity limits the model's potential to internalize new knowledge, eventually leading to performance saturation.

As illustrated in Figure 15, these findings reinforce our primary claim: effective MoE model growth requires a synergistic scaling of both architectural capacity and computational budget.

# D. Discussion on Function Preserving

We observed that, under our model architecture, directly growing a smaller model into a larger one does not lead to severe accuracy degradation on downstream evaluations, even though the outputs for identical inputs may differ due to manual alterations of model weights. Furthermore, the accuracy drop tends to be smaller for width growth compared to depth growth. This phenomenon relates to a principle in model growth known as **Function Preserving** (FP) (Evci et al., 2022; Wang et al., 2023; Yao et al., 2024). FP stipulates that, for any given input, the output before and after model growth should remain identical, thereby guaranteeing that performance is not immediately harmed: $y_{\text{original}}(x) = y_{\text{growth}}(x)$. In practice, however, we find that even when FP rules are not strictly enforced, performance degradation is minor. This robustness can be attributed to the pre-norm structure widely adopted in modern transformers. In **pre-norm** layers, the normalization is applied *before* the residual connection, i.e.,

$$h^{(l+1)} = h^{(l)} + \mathcal{F}\big(\text{LN}(h^{(l)})\big), \tag{4}$$

where $h^{(l)}$ is the input to layer $l$, LN denotes layer normalization, and $\mathcal{F}$ represents the sublayer transformation (e.g., attention or feedforward block).

By contrast, in the original Transformer and BERT, the **post-norm** structure was used, where normalization is applied *after* the residual connection:

$$h^{(l+1)} = \text{LN}\big(h^{(l)} + \mathcal{F}(h^{(l)})\big). \tag{5}$$

Although post-norm structures can better exploit model capacity, they are known to be harder to optimize and less stable during training. Pre-norm designs, in contrast, are easier to train but may reduce the model's effective depth.

This structural distinction explains our empirical findings. Under the pre-norm structure, when layers are duplicated during depth growth, the residual-normalization combination in Equation (4) ensures that the difference between the output of a single layer and that of a duplicated pair of layers is small. As a result, the overall model output remains similar, and performance degradation is limited. In contrast, with post-norm (Equation (5)), duplicating layers alters the scale of normalized outputs more substantially, leading to larger deviations and thus greater performance drops immediately after growth.

Experimental evidence supporting this claim is presented in Figure 16, where we find that evaluating a checkpoint immediately after width growth (before any further training) results in only a minor decrease in downstream task accuracy, or in some cases even a slight improvement due to the inherent randomness of evaluation. In contrast, depth growth can disrupt the functional role of layers, and in older post-layer normalization (Post-LN) architectures like BERT and original Transformer, this would cause a significant performance degradation immediately after expansion. Width growth, however, effectively preserves performance post-expansion in both Pre-LN and Post-LN architectures.

From another perspective, width growth is naturally more function-preserving. When adding experts in MoE layers, both expert weights and router weights are copied. This implies that, at inference time, the widened MoE produces outputs identical to the original configuration, fully consistent with the FP principle. In practice, we add small Gaussian noise to the new experts to encourage specialization during continued training, but such noise

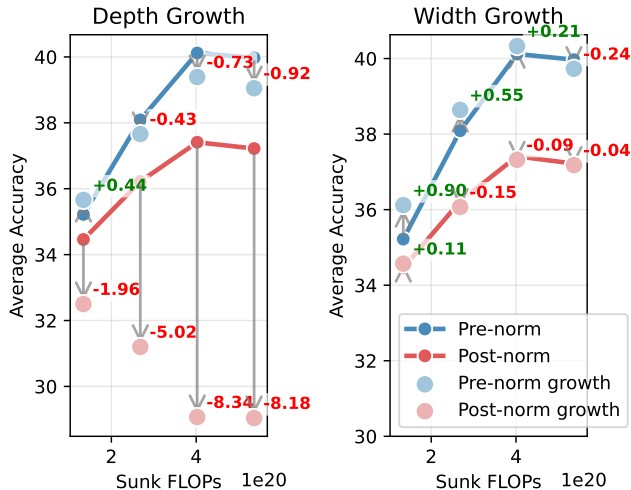

*Figure 16.* Experimental results on Function-Preserving principle between depth and width growth.

only causes negligible shifts in model outputs. Importantly, since width growth operates solely within the MoE module and

does not alter the layer structure, it maintains performance under both pre-norm and post-norm settings. This explains why width growth yields better immediate performance retention than depth growth, as shown in Figure 16.

## E. Orthogonality Analysis

This section provides a comprehensive empirical analysis of the orthogonality between depth and width growth, complementing the summary in Section 3.3. We examine orthogonality from four perspectives: (1) structural disjointness of new parameters, (2) gradient-direction analysis using Adam optimizer states, (3) cumulative weight-delta divergence on shared parameters, and (4) order-independence verification at 12B scale. Both grown models (6B-depth and 6B-width) are initialized from the same base 3B checkpoint and trained on identical data. We extract Adam optimizer states and model weights at multiple post-growth checkpoints for analysis.

### E.1. Structural Orthogonality

Depth growth and width growth operate on **structurally disjoint** regions of the parameter space. Depth growth introduces 3,280 new tensors (16 interposed transformer layers), while width growth introduces 3,840 new tensors (64 new experts per layer × 20 layers). These two parameter sets have **zero intersection**: depth growth does not modify any expert within existing layers, and width growth does not add any new layers.

### E.2. Gradient Direction Analysis

To analyze orthogonality at the optimization level, we extract the Adam optimizer's first moment $m_t = \beta_1 m_{t-1} + (1-\beta_1)g_t$ from training checkpoints at every 2k steps after growth. This exponential moving average of gradients represents the effective optimization direction. We restrict analysis to **shared parameters**, i.e., the parameter tensors across 20 base layers that exist in all three models (base, depth-grown, width-grown). For the depth-grown model, we remap layer indices to the base model's indexing, excluding new layers. For the width-grown model, we truncate expanded expert dimensions to match the original 64 experts.

**Growth redirects optimization orthogonally to the pre-growth direction.** As shown in Figure 18(a) and Table 6, both growth operations redirect the optimization direction to be nearly orthogonal to the pre-growth gradient ($|\cos| < 0.04$) throughout 16k steps of continued training. This indicates that growth operations introduce genuinely new learning signals rather than continuing the pre-growth trajectory.

*Table 6.* Cosine similarity between post-growth and pre-growth gradient directions on shared parameters.

| Post-Growth Steps | $\cos(m_{\text{base}}, m_{\text{depth}})$ | $\cos(m_{\text{base}}, m_{\text{width}})$ |
|---|---|---|
| +2k | 0.031 | −0.003 |
| +4k | 0.039 | 0.022 |
| +6k | 0.012 | 0.034 |
| +8k | −0.001 | 0.027 |
| +10k | 0.012 | 0.001 |
| +12k | −0.009 | −0.018 |
| +14k | −0.008 | 0.011 |
| +16k | 0.016 | 0.038 |

**Cross-growth gradient correlation decreases on MoE-specific parameters.** Table 7 shows the gradient correlation between depth-grown and width-grown models by parameter category. The MoE-specific components (expert FFN and router) exhibit a clear decreasing trend (expert FFN: 0.510→0.392; router: 0.362→0.324), indicating that depth and width growth progressively specialize in different optimization directions on the parameters most central to MoE functionality.

*Table 7.* Cross-growth gradient cosine similarity by parameter category.

| Steps | Overall | Expert FFN | Attention | Router | Embedding |
|---|---|---|---|---|---|
| +2k | 0.480 | 0.510 | 0.380 | 0.362 | 0.835 |
| +8k | 0.610 | 0.497 | 0.509 | 0.404 | 0.894 |
| +16k | 0.585 | **0.392** | 0.469 | **0.324** | 0.854 |

**Per-layer analysis reveals position-dependent specialization.** As shown in Figure 17, a clear layer-position effect emerges at +16k steps: earlier layers (0–9, mean $\cos = 0.342$) show substantially lower gradient correlation than later layers (10–19, mean $\cos = 0.560$). This aligns with the structure of depth growth via interposition, which inserts new layers predominantly in the lower-to-middle range of the network, creating more differentiated gradient flows in earlier layers.

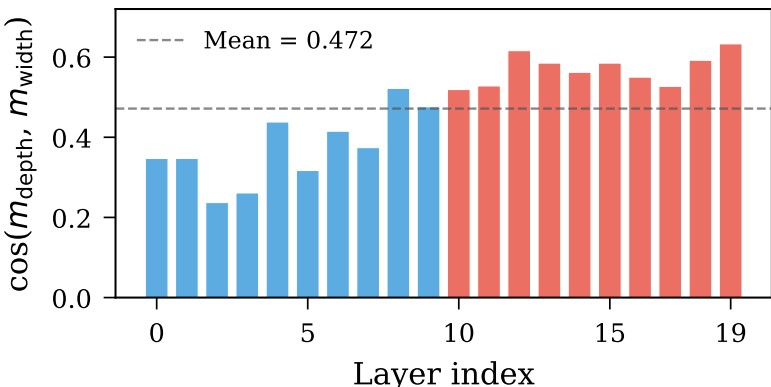

*Figure 17.* Per-layer gradient correlation between depth and width growth at +16k steps. Earlier layers exhibit lower correlation due to the structural impact of interposition.

### E.3. Cumulative Weight Update Divergence

Complementing the gradient-direction analysis, we examine the cosine similarity of cumulative weight changes $\cos(\Delta_\text{depth}, \Delta_\text{width})$ where $\Delta = \theta_\text{grown}(t) - \theta_\text{base}$ on shared parameters. Table 8 reports fine-grained results at every 2k steps from +2k to +20k.

*Table 8.* Cosine similarity of cumulative weight updates between depth-grown and width-grown models on shared parameters.

| Steps | Overall | Expert FFN | Attention | Embedding |
|-------|---------|------------|-----------|-----------|
| +2k   | 0.630   | 0.614      | 0.635     | 0.781     |
| +4k   | 0.564   | 0.547      | 0.599     | 0.750     |
| +6k   | 0.521   | 0.502      | 0.577     | 0.734     |
| +8k   | 0.490   | 0.469      | 0.562     | 0.725     |
| +10k  | 0.468   | 0.446      | 0.556     | 0.720     |
| +12k  | 0.451   | 0.428      | 0.551     | 0.717     |
| +14k  | 0.439   | 0.415      | 0.549     | 0.714     |
| +16k  | 0.431   | 0.406      | 0.551     | 0.713     |
| +18k  | 0.424   | 0.399      | 0.555     | 0.712     |
| +20k  | 0.418   | 0.393      | 0.554     | 0.710     |

The overall weight-delta cosine similarity **monotonically decreases** from 0.63 to 0.42 over 20k steps of training (Figure 18(b)). The initially moderate correlation at +2k reflects a brief recovery phase immediately after architectural perturbation, where both models adjust toward nearby loss minima producing correlated updates. As training progresses, each growth type develops specialized representations, and the weight updates become increasingly decorrelated. Expert FFN parameters (the components most central to MoE functionality) exhibit the strongest divergence (0.614→0.393), while embedding parameters (shared vocabulary representations) maintain higher correlation as expected.

### E.4. Order Independence at 12B Scale

To verify orthogonality at larger scale, we compare the weights of 12B models grown via two different orderings (depth→width and width→depth) at every 1k iteration from 42k to 54k. Both models reach 12B parameters through sequential application of both growth operations but in opposite order.

As shown in Figure 18(c), at iteration 42k (when both models have just completed their second growth), the overall weight cosine similarity is 0.930. Over 12k further training steps, the similarity gradually decreases to 0.823 as the two models

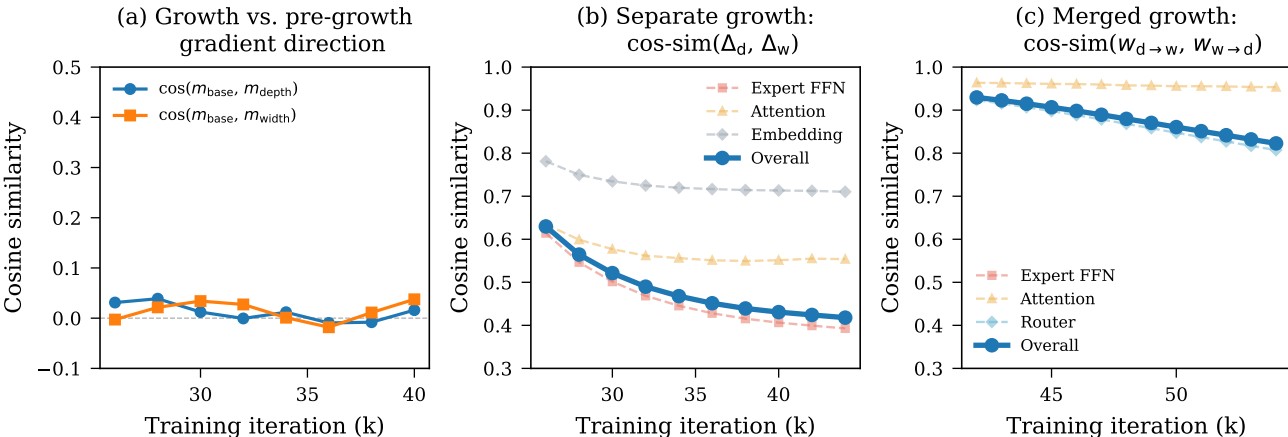

*Figure 18.* Three perspectives on orthogonality between depth and width growth. **(a)** Post-growth gradient directions are nearly orthogonal to the pre-growth direction ($\cos \approx 0$), stable across 16k training steps. **(b)** When grown separately from the same base, the weight updates of depth-grown and width-grown models diverge over training (overall cosine: $0.63 \rightarrow 0.42$). **(c)** When grown sequentially into 12B, the two orderings ($d \rightarrow w$ vs. $w \rightarrow d$) converge to similar solutions (overall cosine $> 0.82$ throughout). Panels (b) and (c) share the same y-axis range to highlight the contrast.

develop slightly different representations during independent training. Crucially, **the similarity remains high ($>$0.82) throughout**, confirming that growth order does not significantly bias the final solution.

Per-category breakdown at iteration 54k reveals that attention parameters show the highest agreement (0.953), followed by embedding (0.901), expert FFN (0.823), and router parameters (0.807). The router showing the lowest similarity is expected, as the routing function is most sensitive to expert composition, which differs structurally between the two growth orderings. The uniform distribution across all 36 layers (range: 0.78–0.86) further confirms that neither growth ordering creates localized disruptions.

### E.5. Summary

Taken together, the above analyses provide converging evidence for practical orthogonality between depth and width growth. Structurally, the two operations expand non-overlapping parameter regions. At the gradient level, each growth redirects optimization nearly orthogonally to the pre-growth direction ($\cos \approx 0$), and the cross-growth gradient correlation on MoE-specific parameters decreases over training. At the weight level, cumulative parameter updates from depth and width growth become increasingly decorrelated (overall cosine: $0.63 \rightarrow 0.42$). At model scale, growth order does not significantly affect the final 12B model (cosine $> 0.82$ throughout). These findings justify the compositional growth strategy proposed in the main paper: because the two growth operations act on largely independent dimensions of both the parameter space and the optimization landscape, they can be applied sequentially in either order with minimal interference.

## F. Detailed Training Settings

This appendix provides details of our pretraining pipeline, including model architecture (Appendix F.1), dataset composition (Appendix F.2), training hyperparameters (Appendix F.3), and infrastructure configurations (Appendix F.4).

### F.1. Model Structure

We adopt a standard decoder-only LLM architecture, with each layer containing Grouped Query Attention (GQA) and Mixture-of-Experts (MoE) modules in both the 3B and 17B models. RMSNorm is used for layer normalization, and rotary position embeddings are applied.

For the 3B model, we set the number of layers to 20 with a hidden size of 1024. The GQA module uses 16 attention heads grouped into 4 query groups. The MoE module consists of 64 experts, of which 4 are activated during computation. The hidden size of each expert is 768.

For the 17B model, we use 28 layers with a hidden size of 2048. GQA again uses 16 attention heads with 4 query groups.

The MoE module includes 96 experts, with 6 activated during computation. Each expert has a hidden size of 1024.

Detailed parameter counts, including both total and activated parameters for the original 3B/17B models and their variants, are provided in Table 9. In depth growth, doubling the number of layers precisely doubles the core parameters (i.e., excluding embedding layers and the LM head). While this linear relationship may not be immediately apparent when comparing the 3.5B/A700M base model with the 6.6B/A1B depth-grown variant, it is numerically consistent when restricted to core weights, as detailed in Table 9. Conversely, in width growth, core parameters expand by less than $2\times$ because only the experts and router weights are duplicated, while the attention mechanisms remain unchanged.

*Table 9.* Detailed parameter counts comparison between various sizes of base and growth models

| Models | Layers | N_expert/active | Total | Active | Core Total* | Core Active* |
|---|---|---|---|---|---|---|
| 3B Base | 20 | 64 / 4 | 3,535,845,376 (3.5B) | 703,461,376 (703M) | 3,126,114,304 | 293,730,304 |
| 6B Width | 20 | 128 / 8 | 6,557,054,976 (6.5B) | 892,286,976 (892M) | 6,147,323,904 | 482,555,904 |
| 6B Depth | 40 | 64 / 4 | 6,661,958,656 (6.6B) | 997,190,656 (1.0B) | 6,252,227,584 | 587,459,584 |
| 12B Double | 40 | 128 / 8 | 12,704,377,856 (12.7B) | 1,374,841,856 (1.4B) | 12,294,646,784 | 965,110,784 |
| 17B Base | 28 | 96 / 6 | 18,030,126,080 (18.0B) | 2,170,496,000 (2.1B) | 17,620,395,008 | 1,760,764,928 |
| 35B Width | 28 | 192 / 12 | 34,947,064,832 (34.9B) | 3,227,804,672 (3.2B) | 34,537,333,760 | 2,818,073,600 |
| 35B Depth | 56 | 96 / 6 | 35,240,787,968 (35.2B) | 3,521,527,808 (3.5B) | 34,831,056,896 | 3,111,796,736 |
| 70B Double | 56 | 192 / 12 | 69,074,665,472 (69.0B) | 5,636,145,152 (5.6B) | 68,664,934,400 | 5,226,414,080 |

*: **Core** means excluding parameter counts in Embedding layer and LM Head (204,865,536 each)

For MoE models specifically, we apply a sigmoid function to compute router scores instead of the softmax function. In the router, expert bias is disabled for the 3B model but enabled for the 17B model. For load balancing, we use sequence-level auxiliary loss in the 3B model and global-batch auxiliary loss in the 17B model.

### F.2. Dataset Composition

Our pretraining corpus is constructed from a diverse and high-quality dataset comprising a mixture of public and proprietary sources, including:

- **DCLM**: dataset released by Apple (Li et al., 2024) with de-duplication (1T tokens)

- **FineWeb-Edu**: dataset released by Hugging Face (Penedo et al., 2024) with de-duplication (280B tokens)

- **Nemotron-CC-HQ**: a high-quality Common Crawl–based dataset (4.67T tokens) released by NVIDIA (Su et al., 2024)

- **Filtered Code Data**: a curated code dataset (640B tokens)

- **Synthetic Data**: high-quality, instruction-oriented synthetic corpora (1.8T tokens)

We randomly shuffle these corpora and uniformly sample approximately 1T tokens for training. We preprocess the raw dataset using the GPT-4o tokenizer, which has a vocabulary size of 200,019. The maximum sequence length is fixed at 4096 tokens. During training, the batch size is set to 1024 for the 3B model and 4096 for the 17B model.

### F.3. Training Hyperparameters

Both for 3B model and 17B model, all learnable parameters are randomly initialized with a standard deviation of 0.02. We employ the AdamW optimizer (Loshchilov & Hutter, 2019) with hyperparameters set to $\beta_1 = 0.9, \beta_2 = 0.95$, and weight-decay = 0.1. Max learning rate is $3 \times 10^{-4}$ for 3B model and $2.6 \times 10^{-4}$ for 17B model. As for the learning rate scheduling, we first linearly increase it from 0 to max learning rate during the first 3K steps. Then, we keep a constant learning rate. For 3B model, we decay it into the minimum learning rate, which is 1/10 of max learning rate during the annealing process. We do not do annealing for 17B model yet. Learning rate is maintained constant at $2.6 \times 10^{-4}$ after warm-up phase.

## F.4. Infrastructure Details

We train our model with mixed precision framework (BF16 + FP32). We use Flash Attention(Dao et al., 2022) for training acceleration. A distributed optimizer is employed to partition optimizer states across data-parallel GPUs, thereby reducing memory consumption. MoE layer recomputation is enabled to further decrease memory usage, and Grouped GeMM (General Matrix Multiplication) is used to accelerate MoE computations. For the 17B model, we use an expert parallel size of 8 to distribute expert weights across GPUs, which allows us to fit within the memory constraints of each device. For infrastructural reasons, we occasionally enable pipeline parallelism (size = 2) to free memory for larger microbatch sizes, improving GeMM efficiency. For the smaller 3B model, we use an expert parallel size of 2 without pipeline parallelism, since the cost of all-to-all expert communication is lower than the overhead introduced by pipeline scheduling and idle bubbles.

# G. Evaluation Details

## G.1. Method for Computing Average Accuracy

We conduct our evaluation through the widely used lm-evaluation-harness library[5] (Gao et al., 2024). All reported average accuracy values in the main text are derived from the average accuracy of following two categories: (1) comprehensive knowledge and reasoning ability, and (2) basic multiple-choice QA performance.

For comprehensive knowledge and reasoning ability, we use the MMLU benchmark (Massive Multitask Language Understanding) (Hendrycks et al., 2020), which consists of 57 tasks spanning STEM, humanities, social sciences, and professional domains. MMLU is widely recognized for assessing models' ability to apply world knowledge, solve problems, and perform reasoning beyond surface-level pattern recognition. We evaluate using a few-shot setting with 5 in-context examples.

In addition, we assess performance on multiple-choice QA benchmarks including ARC (Clark et al., 2018), BoolQ (Clark et al., 2019), HellaSwag (Zellers et al., 2019), LogiQA (Liu et al., 2021), OpenBookQA (ObQA) (Mihaylov et al., 2018), and Winogrande (Sakaguchi et al., 2021). These tasks are evaluated in the zero-shot setting, with accuracy (percentage of correctly chosen options) as the evaluation metric. Collectively, they complement MMLU by emphasizing commonsense and scientific reasoning in narrower but challenging domains.

## G.2. Detailed Evaluation Results

We provide the complete accuracy tables from Table 10 to Table 41. The results presented in the main text as averaged figures or tables are derived directly from these original tables. For clarity, we indicate the corresponding appearances of each result in the table headers.

---

[5]https://github.com/EleutherAI/lm-evaluation-harness

*Table 10.* Full evaluation results of 3B model pretraining in Figure 3, Figure 4 and Figure 7

| Steps | MMLU | QA_average | Arc_C | BoolQ | Hellaswag | Logiqa | Openbookqa | Winogrande | **Average** |
|---|---|---|---|---|---|---|---|---|---|
| 2k | 25.54 | 34.75 | 22.87 | 54.34 | 27.60 | 28.88 | 24.20 | 50.59 | 30.14 |
| 4k | 27.69 | 37.91 | 28.75 | 49.79 | 35.41 | 28.88 | 30.40 | 54.22 | 32.80 |
| 6k | 28.94 | 38.76 | 32.85 | 46.30 | 39.95 | 29.19 | 31.60 | 52.64 | 33.85 |
| 8k | 30.70 | 41.26 | 34.56 | 53.36 | 43.62 | 29.65 | 32.60 | 53.75 | 35.98 |
| 10k | 29.08 | 42.02 | 35.84 | 55.32 | 45.24 | 28.42 | 33.40 | 53.91 | 35.55 |
| 12k | 31.75 | 42.95 | 34.39 | 57.49 | 47.51 | 29.49 | 33.40 | 55.41 | 37.35 |
| 14k | 32.46 | 42.42 | 37.12 | 48.93 | 48.85 | 29.19 | 34.20 | 56.20 | 37.44 |
| 16k | 33.34 | 44.10 | 37.03 | 56.79 | 50.15 | 29.65 | 34.40 | 56.59 | 38.72 |
| 18k | 32.93 | 43.87 | 39.85 | 52.54 | 50.69 | 30.57 | 34.00 | 55.56 | 38.40 |
| 20k | 33.46 | 43.04 | 39.33 | 49.08 | 51.66 | 28.73 | 34.80 | 54.62 | 38.25 |
| 22k | 32.90 | 43.26 | 38.48 | 51.01 | 52.21 | 27.04 | 34.80 | 56.04 | 38.08 |
| 24k | 35.83 | 44.42 | 36.69 | 57.95 | 52.90 | 29.19 | 33.80 | 55.96 | 40.12 |
| 26k | 33.94 | 43.74 | 39.68 | 47.25 | 53.34 | 28.73 | 35.20 | 58.25 | 38.84 |
| 28k | 36.26 | 43.70 | 41.04 | 45.50 | 53.73 | 29.49 | 35.00 | 57.46 | 39.98 |
| 30k | 35.98 | 44.26 | 40.70 | 47.80 | 54.99 | 28.42 | 35.00 | 58.64 | 40.12 |
| 32k | 35.94 | 43.73 | 40.61 | 44.13 | 54.87 | 31.18 | 33.80 | 57.77 | 39.83 |
| 34k | 36.88 | 45.43 | 43.09 | 52.69 | 55.18 | 29.34 | 34.80 | 57.46 | 41.15 |
| 36k | 37.73 | 46.88 | 42.32 | 63.15 | 55.44 | 28.26 | 34.80 | 57.30 | 42.30 |
| 38k | 37.49 | 44.70 | 41.30 | 53.91 | 55.50 | 27.65 | 33.40 | 56.43 | 41.09 |
| 40k | 37.57 | 45.19 | 41.04 | 51.59 | 56.24 | 26.42 | 35.40 | 60.46 | 41.38 |
| 42k | 38.13 | 47.47 | 41.72 | 62.51 | 56.30 | 28.57 | 35.00 | 60.69 | 42.80 |
| 44k | 39.58 | 44.26 | 41.13 | 48.35 | 56.65 | 27.96 | 33.20 | 58.25 | 41.92 |
| 46k | 38.29 | 45.78 | 41.98 | 55.05 | 56.67 | 27.96 | 34.00 | 59.04 | 42.04 |
| 48k | 40.44 | 46.17 | 41.04 | 55.60 | 56.47 | 29.19 | 34.80 | 59.91 | 43.30 |
| 50k | 40.38 | 45.11 | 41.72 | 49.20 | 57.23 | 27.65 | 35.20 | 59.67 | 42.75 |
| 52k | 40.01 | 46.52 | 42.58 | 54.28 | 57.48 | 29.49 | 35.80 | 59.51 | 43.27 |
| 54k | 40.23 | 45.49 | 41.47 | 50.40 | 57.06 | 28.88 | 35.20 | 59.91 | 42.86 |
| 56k | 41.21 | 44.79 | 42.06 | 44.10 | 58.06 | 29.03 | 34.80 | 60.69 | 43.00 |
| 58k | 40.17 | 46.72 | 42.66 | 56.02 | 57.40 | 28.42 | 34.80 | 61.01 | 43.44 |
| 60k | 41.31 | 47.57 | 40.96 | 59.11 | 57.63 | 29.34 | 37.60 | 60.77 | 44.44 |
| 62k | 40.45 | 46.34 | 42.83 | 54.80 | 57.84 | 28.42 | 35.20 | 58.96 | 43.40 |
| 64k | 41.85 | 47.69 | 41.89 | 59.27 | 58.36 | 28.73 | 34.80 | 63.06 | 44.77 |
| 66k | 42.28 | 47.72 | 42.41 | 58.38 | 58.11 | 29.19 | 36.40 | 61.80 | 45.00 |
| 68k | 41.08 | 47.23 | 40.87 | 60.15 | 58.15 | 28.73 | 35.20 | 60.30 | 44.16 |
| 70k | 43.06 | 48.20 | 42.06 | 62.81 | 58.75 | 29.95 | 35.00 | 60.62 | 45.63 |
| 72k | 42.06 | 47.81 | 42.66 | 61.01 | 58.33 | 28.88 | 34.80 | 61.17 | 44.93 |
| 74k | 40.81 | 47.76 | 43.86 | 59.14 | 58.92 | 30.26 | 34.60 | 59.75 | 44.28 |
| 76k | 43.17 | 47.92 | 42.58 | 61.28 | 59.63 | 28.57 | 33.40 | 62.04 | 45.54 |
| 78k | 42.57 | 48.53 | 43.60 | 61.07 | 59.58 | 29.34 | 36.20 | 61.40 | 45.55 |
| 80k | 42.91 | 49.05 | 42.92 | 63.70 | 60.17 | 28.88 | 36.20 | 62.43 | 45.98 |
| 82k | 43.83 | 48.79 | 42.92 | 61.62 | 60.46 | 29.65 | 35.60 | 62.51 | 46.31 |
| 84k | 44.09 | 49.68 | 42.83 | 64.25 | 60.65 | 30.26 | 36.60 | 63.46 | 46.88 |
| 86k | 45.25 | 49.12 | 43.00 | 60.40 | 61.18 | 30.41 | 36.60 | 63.14 | 47.19 |
| 88k | 45.34 | 50.47 | 44.20 | 65.17 | 62.04 | 30.88 | 37.40 | 63.14 | 47.91 |
| 90k | 45.93 | 50.46 | 44.62 | 65.50 | 62.14 | 29.65 | 38.20 | 62.67 | 48.20 |
| 92k | 45.78 | 50.18 | 43.94 | 64.07 | 62.23 | 29.95 | 37.60 | 63.30 | 47.98 |
| 94k | 45.93 | 50.01 | 43.34 | 64.07 | 62.33 | 28.88 | 38.00 | 63.46 | 47.97 |
| 96k | 46.05 | 50.07 | 43.94 | 62.54 | 62.66 | 30.26 | 37.80 | 63.22 | 48.06 |
| 98k | 45.30 | 50.12 | 44.20 | 65.32 | 62.58 | 28.88 | 36.60 | 63.14 | 47.71 |

*Table 11.* Full evaluation results of 6B model pretraining in Figure 7, Figure 8 and Figure 10

| Steps | MMLU | QA_average | Arc_C | BoolQ | Hellaswag | Logiqa | Openbookqa | Winogrande | **Average** |
|---|---|---|---|---|---|---|---|---|---|
| 2k | 24.65 | 36.53 | 23.55 | 60.06 | 28.27 | 27.50 | 28.00 | 51.78 | 30.59 |
| 4k | 27.03 | 38.87 | 29.95 | 59.79 | 37.24 | 27.96 | 27.60 | 50.67 | 32.95 |
| 6k | 30.12 | 39.68 | 32.85 | 50.24 | 42.74 | 27.65 | 32.80 | 51.78 | 34.90 |
| 8k | 31.76 | 42.20 | 36.52 | 52.35 | 47.67 | 29.49 | 32.40 | 54.78 | 36.98 |
| 10k | 30.73 | 44.68 | 38.65 | 61.47 | 50.44 | 28.73 | 32.60 | 56.20 | 37.71 |
| 12k | 33.63 | 44.83 | 38.91 | 59.08 | 51.98 | 28.11 | 34.80 | 56.12 | 39.23 |
| 14k | 33.07 | 45.23 | 41.30 | 56.73 | 53.21 | 27.96 | 34.80 | 57.38 | 39.15 |
| 16k | 32.82 | 44.77 | 40.78 | 51.62 | 55.11 | 26.42 | 35.80 | 58.88 | 38.79 |
| 18k | 36.58 | 46.53 | 40.96 | 59.42 | 55.21 | 28.88 | 36.40 | 58.33 | 41.56 |
| 20k | 35.78 | 43.18 | 41.30 | 40.86 | 56.51 | 27.80 | 35.80 | 56.83 | 39.48 |
| 22k | 38.06 | 45.40 | 42.24 | 48.47 | 57.30 | 29.34 | 35.20 | 59.83 | 41.73 |
| 24k | 39.20 | 46.05 | 42.32 | 54.19 | 57.59 | 26.42 | 37.00 | 58.80 | 42.63 |
| 26k | 38.11 | 45.80 | 42.41 | 49.91 | 58.19 | 28.42 | 35.80 | 60.06 | 41.95 |
| 28k | 40.93 | 45.01 | 40.53 | 46.27 | 59.00 | 28.42 | 36.40 | 59.43 | 42.97 |
| 30k | 42.24 | 45.79 | 43.60 | 43.21 | 59.41 | 28.42 | 38.20 | 61.88 | 44.01 |
| 32k | 42.28 | 47.92 | 44.37 | 56.39 | 59.59 | 29.49 | 37.20 | 60.46 | 45.10 |
| 34k | 42.10 | 46.44 | 44.20 | 48.84 | 59.96 | 28.42 | 38.60 | 58.64 | 44.27 |
| 36k | 43.63 | 46.43 | 44.88 | 44.74 | 60.47 | 29.34 | 38.00 | 61.17 | 45.03 |
| 38k | 42.81 | 47.64 | 45.73 | 48.41 | 61.11 | 29.03 | 38.00 | 63.54 | 45.22 |
| 40k | 44.02 | 47.77 | 45.65 | 51.56 | 60.83 | 28.88 | 39.40 | 60.30 | 45.90 |
| 42k | 44.30 | 49.44 | 45.56 | 62.35 | 61.41 | 29.03 | 37.80 | 60.46 | 46.87 |
| 44k | 44.59 | 47.44 | 45.31 | 48.41 | 61.66 | 29.19 | 38.40 | 61.64 | 46.01 |
| 46k | 44.51 | 47.27 | 46.50 | 46.51 | 61.81 | 28.26 | 37.40 | 63.14 | 45.89 |

*Table 12.* Full evaluation results of 6B model interpositional growth at 2k steps in Figure 4

| Steps | MMLU | QA_average | Arc_C | BoolQ | Hellaswag | Logiqa | Openbookqa | Winogrande | **Average** |
|---|---|---|---|---|---|---|---|---|---|
| 2k | 25.45 | 34.79 | 23.81 | 54.31 | 27.55 | 26.27 | 27.20 | 49.57 | 30.12 |
| 4k | 27.33 | 38.97 | 29.86 | 57.00 | 37.14 | 27.34 | 30.60 | 51.85 | 33.15 |
| 6k | 29.96 | 40.58 | 32.85 | 53.39 | 43.43 | 28.11 | 31.40 | 54.30 | 35.27 |
| 8k | 31.80 | 40.79 | 35.24 | 45.54 | 47.47 | 28.26 | 33.20 | 55.01 | 36.29 |
| 10k | 31.09 | 42.04 | 36.26 | 53.03 | 49.68 | 26.27 | 33.40 | 53.59 | 36.56 |
| 12k | 32.30 | 43.22 | 40.36 | 46.36 | 51.68 | 29.95 | 34.60 | 56.35 | 37.76 |
| 14k | 35.33 | 43.70 | 40.10 | 46.61 | 53.74 | 30.72 | 34.80 | 56.20 | 39.51 |
| 16k | 37.27 | 44.18 | 40.78 | 45.87 | 54.90 | 29.65 | 36.00 | 57.85 | 40.72 |

*Table 13.* Full evaluation results of 6B model stack growth at 2k steps in Figure 4

| Steps | MMLU | QA_average | Arc_C | BoolQ | Hellaswag | Logiqa | Openbookqa | Winogrande | **Average** |
|---|---|---|---|---|---|---|---|---|---|
| 2k | 24.82 | 35.03 | 24.23 | 53.55 | 27.78 | 29.03 | 25.80 | 49.80 | 29.93 |
| 4k | 26.96 | 36.86 | 30.89 | 41.96 | 37.68 | 26.88 | 29.20 | 54.54 | 31.91 |
| 6k | 30.10 | 40.67 | 33.87 | 51.07 | 43.68 | 28.88 | 32.60 | 53.91 | 35.38 |
| 8k | 31.99 | 42.98 | 37.37 | 57.22 | 47.95 | 28.26 | 33.40 | 53.67 | 37.48 |
| 10k | 32.64 | 44.71 | 38.40 | 62.29 | 50.80 | 28.57 | 33.00 | 55.17 | 38.67 |
| 12k | 32.84 | 44.70 | 39.85 | 55.54 | 52.79 | 29.49 | 33.60 | 56.91 | 38.77 |
| 14k | 33.61 | 45.96 | 39.93 | 62.97 | 54.07 | 28.57 | 34.40 | 55.80 | 39.78 |
| 16k | 34.13 | 46.03 | 40.02 | 57.89 | 55.60 | 29.80 | 35.00 | 57.85 | 40.08 |

*Table 14.* Full evaluation results of 6B model interpositional growth at 4k steps in Figure 4

| Steps | MMLU | QA_average | Arc_C | BoolQ | Hellaswag | Logiqa | Openbookqa | Winogrande | **Average** |
|---|---|---|---|---|---|---|---|---|---|
| 4k | 26.48 | 36.96 | 28.16 | 52.35 | 35.37 | 25.50 | 28.60 | 51.78 | 31.72 |
| 6k | 28.38 | 39.37 | 33.87 | 48.07 | 41.73 | 27.34 | 32.00 | 53.20 | 33.87 |
| 8k | 30.07 | 41.77 | 36.35 | 51.83 | 46.69 | 26.88 | 34.00 | 54.85 | 35.92 |
| 10k | 33.56 | 42.39 | 37.63 | 48.32 | 49.50 | 29.34 | 34.00 | 55.56 | 37.98 |
| 12k | 33.54 | 43.58 | 37.20 | 53.00 | 51.50 | 29.65 | 33.80 | 56.35 | 38.56 |
| 14k | 34.95 | 45.22 | 41.55 | 55.41 | 52.70 | 29.65 | 35.60 | 56.43 | 40.09 |
| 16k | 35.89 | 45.26 | 40.02 | 54.22 | 54.63 | 29.34 | 37.00 | 56.35 | 40.58 |
| 18k | 35.96 | 45.33 | 41.04 | 51.28 | 55.05 | 29.80 | 35.20 | 59.59 | 40.64 |

*Table 15.* Full evaluation results of 6B model stack growth at 4k steps in Figure 4

| Steps | MMLU | QA_average | Arc_C | BoolQ | Hellaswag | Logiqa | Openbookqa | Winogrande | **Average** |
|---|---|---|---|---|---|---|---|---|---|
| 4k | 26.75 | 38.39 | 28.75 | 55.26 | 35.48 | 26.11 | 31.20 | 53.51 | 32.57 |
| 6k | 28.33 | 39.20 | 34.22 | 45.54 | 41.85 | 26.73 | 32.40 | 54.46 | 33.77 |
| 8k | 30.47 | 41.69 | 36.95 | 50.83 | 46.24 | 29.65 | 31.60 | 54.85 | 36.08 |
| 10k | 32.62 | 42.49 | 38.14 | 48.38 | 49.42 | 28.57 | 35.60 | 54.85 | 37.56 |
| 12k | 33.41 | 43.70 | 40.02 | 49.57 | 51.43 | 29.49 | 35.40 | 56.27 | 38.55 |
| 14k | 33.02 | 45.10 | 41.30 | 54.53 | 52.94 | 29.95 | 35.20 | 56.67 | 39.06 |
| 16k | 34.56 | 45.35 | 40.61 | 55.44 | 54.94 | 30.11 | 33.80 | 57.22 | 39.96 |
| 18k | 33.83 | 43.94 | 42.83 | 43.85 | 55.83 | 28.26 | 34.60 | 58.25 | 38.88 |

*Table 16.* Full evaluation results of 6B model interpositional growth at 6k steps in Figure 4

| Steps | MMLU | QA_average | Arc_C | BoolQ | Hellaswag | Logiqa | Openbookqa | Winogrande | **Average** |
|---|---|---|---|---|---|---|---|---|---|
| 6k | 28.09 | 37.84 | 29.86 | 46.91 | 39.99 | 27.04 | 30.20 | 53.04 | 32.97 |
| 8k | 27.29 | 42.05 | 35.92 | 53.27 | 44.47 | 30.72 | 33.40 | 54.54 | 34.67 |
| 10k | 31.32 | 44.10 | 37.37 | 59.69 | 48.27 | 30.72 | 33.40 | 55.17 | 37.71 |
| 12k | 34.18 | 42.83 | 38.05 | 46.82 | 50.77 | 29.95 | 33.60 | 57.77 | 38.50 |
| 14k | 35.29 | 45.84 | 38.82 | 59.94 | 52.73 | 31.34 | 35.20 | 56.99 | 40.56 |
| 16k | 35.06 | 45.63 | 40.87 | 57.95 | 54.06 | 27.34 | 33.60 | 59.98 | 40.35 |
| 18k | 36.72 | 46.90 | 41.13 | 61.80 | 55.34 | 28.88 | 36.00 | 58.25 | 41.81 |
| 20k | 35.66 | 47.10 | 41.04 | 59.17 | 55.91 | 29.80 | 35.40 | 61.25 | 41.38 |

*Table 17.* Full evaluation results of 6B model stack growth at 6k steps in Figure 4

| Steps | MMLU | QA_average | Arc_C | BoolQ | Hellaswag | Logiqa | Openbookqa | Winogrande | **Average** |
|---|---|---|---|---|---|---|---|---|---|
| 6k | 28.11 | 40.17 | 32.68 | 55.50 | 40.37 | 29.03 | 31.40 | 52.01 | 34.14 |
| 8k | 29.23 | 41.90 | 35.32 | 54.19 | 44.66 | 31.03 | 33.60 | 52.57 | 35.56 |
| 10k | 30.17 | 42.47 | 37.88 | 48.32 | 48.48 | 29.34 | 35.00 | 55.80 | 36.32 |
| 12k | 32.57 | 41.98 | 38.31 | 41.41 | 50.88 | 29.19 | 34.80 | 57.30 | 37.28 |
| 14k | 31.79 | 43.15 | 38.99 | 45.63 | 52.42 | 29.95 | 36.20 | 55.72 | 37.47 |
| 16k | 33.08 | 44.19 | 41.04 | 47.77 | 54.11 | 29.19 | 37.00 | 56.04 | 38.64 |
| 18k | 33.64 | 45.11 | 41.98 | 51.56 | 55.29 | 29.03 | 35.20 | 57.62 | 39.38 |
| 20k | 33.56 | 44.23 | 40.96 | 45.20 | 56.21 | 29.65 | 35.80 | 57.54 | 38.89 |

*Table 18.* Full evaluation results of 6B model interpositional growth at 8k steps in Figure 4

| Steps | MMLU | QA_average | Arc_C | BoolQ | Hellaswag | Logiqa | Openbookqa | Winogrande | **Average** |
|---|---|---|---|---|---|---|---|---|---|
| 8k | 28.11 | 40.01 | 32.08 | 53.73 | 43.38 | 27.19 | 31.80 | 51.85 | 34.06 |
| 10k | 28.42 | 42.52 | 37.20 | 51.68 | 46.94 | 30.26 | 34.40 | 54.62 | 35.47 |
| 12k | 32.82 | 43.98 | 37.88 | 52.32 | 49.96 | 31.64 | 34.80 | 57.30 | 38.40 |
| 14k | 35.18 | 44.94 | 40.27 | 55.84 | 52.32 | 30.26 | 34.80 | 56.12 | 40.06 |
| 16k | 35.51 | 46.98 | 39.76 | 61.35 | 53.71 | 31.34 | 37.40 | 58.33 | 41.25 |
| 18k | 36.54 | 46.31 | 41.55 | 53.79 | 54.69 | 30.88 | 35.20 | 61.72 | 41.42 |
| 20k | 37.09 | 46.70 | 42.58 | 57.46 | 56.13 | 29.19 | 36.40 | 58.41 | 41.89 |
| 22k | 37.27 | 45.79 | 42.24 | 48.04 | 56.67 | 29.95 | 36.60 | 61.25 | 41.53 |

*Table 19.* Full evaluation results of 6B model stack growth at 8k steps in Figure 4

| Steps | MMLU | QA_average | Arc_C | BoolQ | Hellaswag | Logiqa | Openbookqa | Winogrande | **Average** |
|---|---|---|---|---|---|---|---|---|---|
| 8k | 30.02 | 41.87 | 34.64 | 56.45 | 43.59 | 29.49 | 32.40 | 54.62 | 35.94 |
| 10k | 28.31 | 42.21 | 36.43 | 48.72 | 46.62 | 30.57 | 36.20 | 54.70 | 35.26 |
| 12k | 32.41 | 44.26 | 39.16 | 53.94 | 50.01 | 30.41 | 35.20 | 56.83 | 38.33 |
| 14k | 33.04 | 44.42 | 39.16 | 52.66 | 52.22 | 29.49 | 36.00 | 56.99 | 38.73 |
| 16k | 33.84 | 45.16 | 38.91 | 59.91 | 53.68 | 29.19 | 34.40 | 54.85 | 39.50 |
| 18k | 33.86 | 45.54 | 40.78 | 52.17 | 54.67 | 31.95 | 35.60 | 58.09 | 39.70 |
| 20k | 34.34 | 44.66 | 42.41 | 49.11 | 55.76 | 27.96 | 35.20 | 57.54 | 39.50 |
| 22k | 34.46 | 45.77 | 41.55 | 51.59 | 56.78 | 30.41 | 37.00 | 57.30 | 40.12 |

*Table 20.* Full evaluation results of 6B model interpositional growth at 24k steps in Figure 3, Figure 8 and Figure 10

| Steps | MMLU | QA_average | Arc_C | BoolQ | Hellaswag | Logiqa | Openbookqa | Winogrande | **Average** |
|---|---|---|---|---|---|---|---|---|---|
| 24k | 34.22 | 44.57 | 37.80 | 62.45 | 50.90 | 27.80 | 33.20 | 55.25 | 39.39 |
| 26k | 35.86 | 45.03 | 39.76 | 52.59 | 54.62 | 30.57 | 35.00 | 57.62 | 40.44 |
| 28k | 36.53 | 45.21 | 40.53 | 52.02 | 55.76 | 29.19 | 34.80 | 58.96 | 40.87 |
| 30k | 39.30 | 45.40 | 41.04 | 51.62 | 56.94 | 28.42 | 35.60 | 58.80 | 42.35 |
| 32k | 40.50 | 45.73 | 42.24 | 51.62 | 57.67 | 29.19 | 35.20 | 58.48 | 43.12 |
| 34k | 41.50 | 46.61 | 43.34 | 53.76 | 57.56 | 29.34 | 35.80 | 59.83 | 44.05 |
| 36k | 42.38 | 48.17 | 42.24 | 62.11 | 58.75 | 27.96 | 36.40 | 61.56 | 45.28 |
| 38k | 43.46 | 48.53 | 44.28 | 61.56 | 58.85 | 28.73 | 37.00 | 60.77 | 46.00 |
| 40k | 42.61 | 48.75 | 43.17 | 63.79 | 59.31 | 28.73 | 35.80 | 61.72 | 45.68 |
| 42k | 44.30 | 49.22 | 45.22 | 65.60 | 59.79 | 28.11 | 36.00 | 60.62 | 46.76 |
| 44k | 44.30 | 48.73 | 46.08 | 58.93 | 60.06 | 30.57 | 37.20 | 59.51 | 46.51 |
| 46k | 45.11 | 48.48 | 44.28 | 56.24 | 60.71 | 30.72 | 36.20 | 62.75 | 46.80 |
| 48k | 46.03 | 49.36 | 44.88 | 62.66 | 60.33 | 28.42 | 36.80 | 63.06 | 47.69 |
| 50k | 44.54 | 49.38 | 43.43 | 61.87 | 61.11 | 30.88 | 37.20 | 61.80 | 46.96 |
| 52k | 46.52 | 49.47 | 45.05 | 57.92 | 61.23 | 30.88 | 38.20 | 63.54 | 48.00 |
| 54k | 46.99 | 48.61 | 46.16 | 51.10 | 62.19 | 30.26 | 39.20 | 62.75 | 47.80 |

*Table 21.* Full evaluation results of 6B model stack growth at 24k steps in Figure 3

| Steps | MMLU | QA_average | Arc_C | BoolQ | Hellaswag | Logiqa | Openbookqa | Winogrande | **Average** |
|-------|------|-----------|-------|-------|-----------|--------|-----------|-----------|---------|
| 24k | 34.14 | 44.94 | 37.20 | 60.24 | 52.35 | 28.57 | 35.40 | 55.88 | 39.54 |
| 26k | 33.69 | 44.83 | 41.21 | 50.73 | 54.61 | 29.95 | 35.00 | 57.46 | 39.26 |
| 28k | 34.77 | 45.31 | 41.38 | 54.65 | 55.60 | 28.88 | 33.40 | 57.93 | 40.04 |
| 30k | 36.41 | 46.14 | 41.30 | 57.55 | 56.67 | 29.03 | 35.00 | 57.30 | 41.28 |
| 32k | 37.54 | 44.92 | 42.92 | 49.17 | 56.88 | 28.57 | 34.20 | 57.77 | 41.23 |
| 34k | 36.68 | 46.71 | 43.09 | 54.62 | 57.86 | 28.57 | 36.20 | 59.91 | 41.69 |
| 36k | 39.20 | 47.42 | 43.26 | 58.13 | 58.34 | 29.19 | 36.40 | 59.19 | 43.31 |
| 38k | 39.70 | 47.61 | 43.34 | 60.55 | 58.92 | 29.65 | 34.00 | 59.19 | 43.65 |
| 40k | 36.92 | 47.39 | 42.75 | 55.96 | 59.40 | 30.72 | 36.00 | 59.51 | 42.16 |

*Table 22.* Full evaluation results of 6B model interpositional growth at 8k steps in Figure 8 and Figure 10

| Steps | MMLU | QA_average | Arc_C | BoolQ | Hellaswag | Logiqa | Openbookqa | Winogrande | **Average** |
|-------|------|-----------|-------|-------|-----------|--------|-----------|-----------|---------|
| 8k | 29.75 | 41.58 | 34.30 | 59.72 | 43.29 | 29.65 | 29.80 | 52.72 | 35.67 |
| 10k | 30.40 | 41.91 | 34.98 | 52.48 | 46.70 | 29.19 | 33.20 | 54.93 | 36.16 |
| 12k | 32.64 | 41.68 | 35.67 | 48.69 | 49.62 | 27.34 | 32.40 | 56.35 | 37.16 |
| 14k | 33.35 | 44.27 | 37.29 | 56.45 | 51.87 | 27.34 | 34.60 | 58.09 | 38.81 |
| 16k | 33.95 | 45.13 | 40.61 | 54.10 | 53.55 | 30.57 | 35.60 | 56.35 | 39.54 |
| 18k | 36.41 | 46.74 | 40.02 | 61.80 | 54.59 | 29.95 | 36.00 | 58.09 | 41.58 |
| 20k | 38.53 | 44.96 | 40.02 | 54.07 | 55.06 | 27.65 | 34.80 | 58.17 | 41.75 |
| 22k | 38.33 | 45.80 | 42.66 | 54.10 | 56.27 | 28.26 | 35.60 | 57.93 | 42.07 |
| 24k | 38.72 | 47.38 | 42.92 | 59.57 | 57.15 | 29.34 | 35.00 | 60.30 | 43.05 |
| 26k | 39.78 | 48.56 | 42.92 | 64.19 | 57.44 | 29.49 | 36.40 | 60.93 | 44.17 |
| 28k | 41.53 | 46.35 | 43.86 | 50.95 | 58.15 | 28.73 | 36.00 | 60.38 | 43.94 |
| 30k | 41.40 | 46.74 | 42.06 | 52.72 | 58.75 | 29.19 | 35.60 | 62.12 | 44.07 |
| 32k | 43.13 | 48.44 | 42.83 | 61.16 | 58.79 | 29.65 | 36.80 | 61.40 | 45.78 |
| 34k | 42.12 | 46.95 | 42.49 | 51.22 | 59.62 | 29.80 | 35.40 | 63.14 | 44.53 |
| 36k | 44.05 | 48.12 | 43.60 | 58.69 | 60.23 | 30.57 | 34.80 | 60.85 | 46.09 |
| 38k | 44.70 | 48.44 | 44.03 | 58.47 | 60.69 | 27.96 | 37.20 | 62.27 | 46.57 |
| 40k | 43.66 | 48.69 | 46.08 | 56.97 | 60.59 | 29.65 | 36.80 | 62.04 | 46.17 |
| 42k | 44.75 | 49.35 | 45.48 | 59.76 | 60.59 | 29.65 | 38.00 | 62.59 | 47.05 |
| 44k | 45.46 | 49.23 | 44.71 | 60.76 | 60.88 | 29.95 | 36.40 | 62.67 | 47.34 |
| 46k | 45.39 | 48.94 | 45.82 | 57.46 | 61.25 | 29.49 | 36.80 | 62.80 | 47.16 |
| 48k | 46.15 | 49.18 | 44.45 | 57.37 | 61.51 | 30.88 | 37.80 | 63.06 | 47.66 |

*Table 23.* Full evaluation results of 6B model interpositional growth at 16k steps in Figure 8 and Figure 10

| Steps | MMLU | QA_average | Arc_C | BoolQ | Hellaswag | Logiqa | Openbookqa | Winogrande | **Average** |
|---|---|---|---|---|---|---|---|---|---|
| 16k | 31.23 | 44.10 | 36.35 | 60.73 | 48.80 | 30.72 | 32.80 | 55.17 | 37.66 |
| 18k | 34.09 | 43.29 | 37.12 | 50.76 | 51.92 | 30.41 | 33.40 | 56.12 | 38.69 |
| 20k | 33.76 | 44.06 | 38.74 | 50.06 | 53.70 | 29.19 | 34.80 | 57.85 | 38.91 |
| 22k | 37.69 | 46.01 | 40.96 | 54.13 | 55.20 | 31.18 | 35.40 | 59.19 | 41.85 |
| 24k | 38.04 | 44.89 | 40.61 | 47.52 | 55.93 | 30.57 | 36.20 | 58.48 | 41.46 |
| 26k | 39.81 | 44.63 | 41.30 | 47.80 | 56.48 | 27.80 | 35.20 | 59.19 | 42.22 |
| 28k | 41.23 | 46.99 | 40.78 | 60.37 | 57.35 | 28.73 | 36.20 | 58.48 | 44.11 |
| 30k | 41.65 | 47.65 | 43.34 | 57.92 | 57.92 | 29.49 | 36.40 | 60.85 | 44.65 |
| 32k | 41.41 | 48.63 | 43.00 | 62.51 | 58.41 | 29.95 | 37.60 | 60.30 | 45.02 |
| 34k | 42.89 | 48.06 | 43.69 | 59.88 | 58.63 | 29.65 | 38.20 | 58.33 | 45.48 |
| 36k | 43.57 | 47.32 | 43.77 | 56.12 | 58.66 | 28.11 | 37.80 | 59.43 | 45.44 |
| 38k | 44.87 | 49.06 | 43.09 | 62.54 | 59.41 | 30.26 | 36.80 | 62.27 | 46.97 |
| 40k | 45.24 | 49.02 | 43.52 | 63.64 | 59.86 | 27.34 | 36.80 | 62.98 | 47.13 |
| 42k | 44.01 | 48.44 | 43.34 | 57.19 | 60.52 | 29.49 | 38.00 | 62.12 | 46.23 |
| 44k | 45.34 | 46.91 | 42.92 | 49.63 | 60.62 | 29.49 | 37.00 | 61.80 | 46.13 |
| 46k | 45.22 | 49.51 | 45.39 | 57.68 | 61.23 | 32.41 | 38.40 | 61.96 | 47.37 |
| 48k | 45.70 | 49.53 | 44.97 | 60.43 | 61.34 | 28.88 | 37.60 | 63.93 | 47.61 |
| 50k | 46.21 | 50.64 | 45.73 | 63.27 | 61.37 | 30.88 | 38.00 | 64.56 | 48.42 |
| 52k | 46.05 | 51.01 | 46.25 | 67.65 | 61.73 | 30.41 | 37.80 | 62.19 | 48.53 |

*Table 24.* Full evaluation results of 6B model interpositional growth at 32k steps in Figure 8 and Figure 10

| Steps | MMLU | QA_average | Arc_C | BoolQ | Hellaswag | Logiqa | Openbookqa | Winogrande | **Average** |
|---|---|---|---|---|---|---|---|---|---|
| 32k | 34.45 | 43.64 | 40.61 | 49.02 | 53.48 | 28.88 | 33.20 | 56.67 | 39.05 |
| 34k | 36.84 | 44.41 | 40.96 | 50.80 | 56.41 | 27.65 | 33.60 | 57.06 | 40.63 |
| 36k | 38.76 | 46.45 | 41.13 | 54.46 | 57.11 | 30.57 | 35.20 | 60.22 | 42.60 |
| 38k | 39.95 | 47.68 | 43.26 | 61.44 | 58.15 | 30.11 | 34.80 | 58.33 | 43.82 |
| 40k | 41.29 | 46.62 | 42.49 | 55.05 | 58.30 | 28.88 | 35.00 | 59.98 | 43.95 |
| 42k | 41.21 | 47.39 | 40.87 | 61.31 | 58.95 | 28.42 | 35.80 | 58.96 | 44.30 |
| 44k | 43.70 | 47.55 | 43.17 | 57.03 | 59.91 | 29.19 | 36.00 | 60.01 | 45.63 |
| 46k | 44.30 | 49.20 | 45.05 | 64.65 | 59.75 | 29.03 | 36.20 | 60.54 | 46.75 |
| 48k | 43.01 | 48.67 | 44.28 | 61.65 | 60.09 | 27.96 | 36.80 | 61.25 | 45.84 |
| 50k | 45.24 | 48.64 | 44.37 | 62.23 | 60.39 | 29.49 | 35.00 | 60.38 | 46.94 |
| 52k | 45.68 | 48.67 | 45.82 | 56.61 | 60.52 | 30.72 | 37.60 | 60.77 | 47.18 |
| 54k | 45.15 | 49.75 | 44.03 | 65.32 | 61.01 | 28.73 | 36.40 | 62.98 | 47.45 |
| 56k | 46.18 | 50.12 | 44.80 | 67.71 | 60.88 | 26.88 | 37.40 | 63.06 | 48.15 |

*Table 25.* Full evaluation results of 6B model interpositional growth at 40k steps in Figure 8 and Figure 10

| Steps | MMLU | QA_average | Arc_C | BoolQ | Hellaswag | Logiqa | Openbookqa | Winogrande | **Average** |
|---|---|---|---|---|---|---|---|---|---|
| 40k | 36.87 | 45.36 | 39.59 | 59.08 | 54.18 | 26.88 | 34.00 | 58.41 | 41.11 |
| 42k | 38.41 | 46.43 | 41.72 | 53.94 | 57.27 | 30.57 | 35.80 | 59.27 | 42.42 |
| 44k | 40.68 | 46.91 | 42.06 | 56.70 | 58.33 | 28.73 | 35.20 | 60.46 | 43.80 |
| 46k | 41.70 | 47.79 | 43.00 | 62.26 | 58.92 | 28.42 | 34.40 | 59.75 | 44.75 |
| 48k | 43.22 | 47.28 | 43.77 | 54.50 | 59.24 | 26.88 | 36.40 | 62.90 | 45.25 |
| 50k | 42.74 | 47.91 | 43.26 | 56.21 | 59.66 | 30.11 | 36.40 | 61.80 | 45.32 |
| 52k | 44.47 | 48.39 | 43.60 | 57.28 | 60.60 | 29.80 | 36.60 | 62.43 | 46.43 |
| 54k | 45.73 | 48.27 | 43.17 | 61.53 | 60.49 | 28.57 | 34.00 | 61.88 | 47.00 |
| 56k | 44.91 | 49.11 | 43.86 | 63.46 | 60.88 | 28.57 | 35.40 | 62.51 | 47.01 |
| 58k | 46.55 | 50.09 | 45.31 | 63.61 | 61.25 | 29.03 | 38.80 | 62.51 | 48.32 |
| 60k | 46.34 | 49.05 | 44.88 | 58.72 | 61.50 | 31.64 | 36.40 | 61.17 | 47.70 |

*Table 26.* Full evaluation results of 6B model interpositional growth at 48k steps in Figure 8 and Figure 10

| Steps | MMLU | QA_average | Arc_C | BoolQ | Hellaswag | Logiqa | Openbookqa | Winogrande | **Average** |
|---|---|---|---|---|---|---|---|---|---|
| 48k | 38.14 | 46.38 | 41.81 | 61.01 | 54.74 | 27.80 | 35.80 | 57.14 | 42.26 |
| 50k | 40.88 | 47.66 | 42.56 | 61.56 | 57.46 | 30.15 | 34.61 | 59.64 | 44.27 |
| 52k | 41.65 | 48.64 | 43.00 | 63.85 | 58.71 | 31.03 | 34.40 | 60.85 | 45.15 |
| 54k | 43.24 | 48.76 | 43.86 | 61.47 | 59.53 | 30.11 | 36.60 | 61.01 | 46.00 |
| 56k | 44.23 | 49.00 | 43.77 | 63.12 | 59.71 | 28.88 | 36.40 | 62.12 | 46.62 |
| 58k | 43.23 | 48.87 | 43.60 | 59.36 | 60.35 | 29.95 | 36.60 | 63.38 | 46.05 |
| 60k | 44.95 | 50.67 | 45.73 | 65.57 | 60.89 | 30.72 | 38.00 | 63.12 | 47.81 |
| 62k | 44.98 | 49.41 | 44.20 | 64.43 | 61.23 | 28.26 | 36.40 | 61.96 | 47.20 |

*Table 27.* Full evaluation results of 6B model interpositional growth at 56k steps in Figure 8 and Figure 10

| Steps | MMLU | QA_average | Arc_C | BoolQ | Hellaswag | Logiqa | Openbookqa | Winogrande | **Average** |
|---|---|---|---|---|---|---|---|---|---|
| 56k | 40.41 | 47.31 | 41.47 | 63.49 | 55.81 | 28.11 | 35.40 | 59.59 | 43.86 |
| 58k | 42.44 | 46.71 | 43.60 | 54.16 | 58.95 | 29.65 | 35.00 | 58.88 | 44.57 |
| 60k | 43.17 | 48.28 | 44.37 | 61.28 | 59.30 | 28.73 | 34.80 | 61.17 | 45.72 |
| 62k | 42.81 | 49.61 | 44.28 | 65.75 | 60.23 | 29.03 | 36.80 | 61.56 | 46.21 |
| 64k | 45.24 | 48.86 | 44.03 | 59.54 | 60.37 | 31.03 | 36.40 | 61.80 | 47.05 |
| 66k | 44.77 | 49.29 | 46.76 | 58.62 | 61.14 | 29.95 | 37.60 | 61.64 | 47.03 |
| 68k | 45.11 | 49.68 | 42.75 | 66.15 | 61.41 | 27.80 | 36.80 | 63.14 | 47.39 |
| 70k | 45.36 | 49.38 | 42.68 | 65.38 | 60.45 | 29.70 | 36.20 | 61.89 | 47.37 |

*Table 28.* Full evaluation results of 6B model interpositional growth at 64k steps in Figure 8 and Figure 10

| Steps | MMLU | QA_average | Arc_C | BoolQ | Hellaswag | Logiqa | Openbookqa | Winogrande | **Average** |
|---|---|---|---|---|---|---|---|---|---|
| 64k | 39.47 | 47.17 | 42.41 | 59.97 | 55.83 | 29.19 | 35.00 | 60.62 | 43.32 |
| 66k | 43.46 | 47.74 | 42.92 | 57.03 | 59.50 | 29.19 | 36.60 | 61.17 | 45.60 |
| 68k | 43.93 | 48.95 | 44.80 | 61.93 | 59.61 | 29.34 | 35.80 | 62.19 | 46.44 |
| 70k | 43.36 | 48.89 | 44.03 | 62.26 | 60.37 | 28.88 | 34.40 | 63.38 | 46.12 |
| 72k | 45.09 | 49.44 | 45.48 | 62.29 | 60.43 | 29.80 | 35.60 | 63.06 | 47.27 |
| 74k | 44.92 | 49.44 | 45.05 | 58.72 | 61.35 | 31.34 | 37.60 | 62.59 | 47.18 |
| 76k | 46.00 | 50.34 | 44.80 | 67.06 | 61.67 | 29.03 | 36.80 | 62.67 | 48.17 |
| 78k | 45.67 | 49.95 | 43.68 | 66.48 | 60.59 | 30.50 | 36.70 | 61.74 | 47.81 |

*Table 29.* Full evaluation results of 6B model interpositional growth at 72k steps in Figure 8

| Steps | MMLU | QA_average | Arc_C | BoolQ | Hellaswag | Logiqa | Openbookqa | Winogrande | **Average** |
|---|---|---|---|---|---|---|---|---|---|
| 72k | 40.44 | 46.88 | 42.24 | 60.52 | 55.79 | 29.03 | 34.40 | 59.27 | 43.66 |
| 74k | 44.35 | 49.47 | 45.56 | 65.17 | 59.75 | 29.95 | 35.40 | 61.01 | 46.91 |
| 76k | 44.97 | 49.50 | 45.31 | 67.28 | 60.02 | 30.11 | 34.00 | 60.30 | 47.24 |
| 78k | 44.45 | 48.94 | 44.45 | 61.68 | 60.89 | 28.57 | 34.80 | 63.22 | 46.69 |
| 80k | 45.33 | 50.27 | 46.08 | 65.05 | 60.88 | 29.80 | 36.20 | 63.61 | 47.80 |
| 82k | 45.27 | 50.97 | 45.05 | 66.79 | 61.67 | 31.34 | 37.60 | 63.38 | 48.12 |
| 84k | 46.61 | 50.76 | 45.14 | 67.92 | 61.83 | 29.49 | 35.80 | 64.40 | 48.69 |
| 86k | 46.32 | 49.48 | 45.14 | 61.16 | 61.91 | 29.03 | 35.40 | 64.25 | 47.90 |

*Table 30.* Full evaluation results of 6B model interpositional growth at 80k steps in Figure 8

| Steps | MMLU | QA_average | Arc_C | BoolQ | Hellaswag | Logiqa | Openbookqa | Winogrande | **Average** |
|---|---|---|---|---|---|---|---|---|---|
| 80k | 41.46 | 48.44 | 41.64 | 65.54 | 57.39 | 28.73 | 36.00 | 61.33 | 44.95 |
| 82k | 44.68 | 49.58 | 44.03 | 62.11 | 60.33 | 30.57 | 37.60 | 62.83 | 47.13 |
| 84k | 45.46 | 49.17 | 44.28 | 63.98 | 60.76 | 29.34 | 36.20 | 60.46 | 47.32 |
| 86k | 43.46 | 49.56 | 45.65 | 64.53 | 61.17 | 28.26 | 35.80 | 61.96 | 46.51 |
| 88k | 46.30 | 49.27 | 44.97 | 61.13 | 61.03 | 27.50 | 36.40 | 64.56 | 47.78 |
| 90k | 45.86 | 50.71 | 44.88 | 67.09 | 61.78 | 29.95 | 37.20 | 63.38 | 48.29 |
| 92k | 46.86 | 50.46 | 44.03 | 65.47 | 62.63 | 29.49 | 37.60 | 63.54 | 48.66 |
| 94k | 47.12 | 50.39 | 45.39 | 65.57 | 62.43 | 28.57 | 36.00 | 64.40 | 48.76 |

*Table 31.* Full evaluation results of 6B model interpositional growth at 88k steps in Figure 8

| Steps | MMLU | QA_average | Arc_C | BoolQ | Hellaswag | Logiqa | Openbookqa | Winogrande | **Average** |
|---|---|---|---|---|---|---|---|---|---|
| 88k | 43.78 | 49.09 | 44.45 | 63.85 | 58.70 | 29.49 | 35.60 | 62.43 | 46.43 |
| 90k | 44.52 | 49.75 | 43.26 | 64.01 | 60.38 | 30.11 | 37.60 | 63.14 | 47.14 |
| 92k | 45.76 | 48.43 | 45.22 | 58.90 | 60.69 | 29.95 | 35.20 | 60.62 | 47.10 |
| 94k | 43.42 | 50.46 | 45.22 | 67.82 | 61.11 | 29.19 | 35.80 | 63.61 | 46.94 |
| 96k | 45.95 | 50.73 | 44.88 | 68.59 | 61.10 | 29.80 | 37.20 | 62.83 | 48.34 |
| 98k | 45.59 | 51.92 | 46.33 | 68.81 | 62.13 | 31.95 | 38.60 | 63.69 | 48.75 |
| 100k | 47.80 | 50.93 | 44.20 | 66.09 | 62.52 | 30.88 | 36.80 | 65.11 | 49.37 |
| 102k | 47.52 | 50.45 | 45.05 | 66.91 | 62.44 | 27.96 | 37.20 | 63.14 | 48.99 |

*Table 32.* Full evaluation results of 6B model interpositional growth at 96k steps in Figure 8

| Steps | MMLU | QA_average | Arc_C | BoolQ | Hellaswag | Logiqa | Openbookqa | Winogrande | **Average** |
|---|---|---|---|---|---|---|---|---|---|
| 96k | 43.80 | 48.58 | 43.52 | 63.58 | 58.94 | 28.57 | 35.60 | 61.25 | 46.19 |
| 98k | 45.05 | 49.03 | 43.77 | 57.49 | 60.76 | 30.72 | 37.60 | 63.85 | 47.04 |
| 100k | 45.28 | 49.44 | 44.54 | 64.59 | 60.84 | 30.57 | 35.00 | 61.09 | 47.36 |
| 102k | 44.49 | 50.30 | 45.56 | 66.57 | 60.69 | 29.95 | 36.60 | 62.43 | 47.40 |
| 104k | 45.94 | 50.56 | 45.56 | 64.68 | 61.45 | 29.65 | 37.60 | 64.40 | 48.25 |
| 106k | 45.88 | 51.12 | 45.56 | 65.93 | 61.55 | 30.57 | 37.60 | 65.51 | 48.50 |
| 108k | 47.41 | 51.11 | 44.80 | 67.89 | 62.31 | 29.19 | 38.00 | 64.48 | 49.26 |
| 110k | 46.96 | 50.08 | 44.71 | 65.66 | 62.25 | 28.57 | 36.00 | 63.30 | 48.52 |

*Table 33.* Full evaluation results of 6B model width growth with no noise in Figure 6

| Steps | MMLU | QA_average | Arc_C | BoolQ | Hellaswag | Logiqa | Openbookqa | Winogrande | **Average** |
|---|---|---|---|---|---|---|---|---|---|
| 24k | 35.85 | 44.65 | 36.86 | 58.10 | 52.79 | 29.34 | 33.80 | 56.99 | 40.25 |
| 26k | 33.89 | 44.02 | 39.51 | 48.10 | 54.52 | 29.49 | 35.60 | 56.87 | 38.95 |
| 28k | 35.22 | 43.14 | 40.61 | 45.78 | 54.67 | 28.26 | 33.40 | 56.12 | 39.18 |
| 30k | 36.35 | 45.00 | 41.04 | 52.39 | 55.99 | 30.57 | 33.40 | 56.59 | 40.67 |
| 32k | 37.60 | 45.61 | 42.49 | 55.02 | 56.11 | 28.26 | 35.60 | 56.20 | 41.61 |
| 34k | 37.56 | 45.98 | 42.41 | 55.26 | 56.61 | 28.42 | 36.80 | 56.35 | 41.77 |
| 36k | 38.38 | 44.37 | 40.78 | 43.84 | 58.02 | 27.80 | 36.40 | 59.35 | 41.37 |
| 38k | 39.83 | 46.66 | 43.26 | 54.89 | 58.00 | 29.34 | 35.20 | 59.27 | 43.25 |
| 40k | 38.78 | 48.59 | 44.62 | 63.12 | 58.79 | 27.96 | 36.40 | 60.62 | 43.68 |
| 42k | 41.30 | 47.69 | 45.48 | 57.40 | 59.16 | 27.96 | 36.60 | 59.51 | 44.49 |
| 44k | 41.35 | 47.65 | 44.88 | 56.73 | 59.37 | 29.80 | 36.40 | 58.72 | 44.50 |

*Table 34.* Full evaluation results of 6B model width growth with noise std=0.01 in Figure 6 and Figure 7

| Steps | MMLU | QA_average | Arc_C | BoolQ | Hellaswag | Logiqa | Openbookqa | Winogrande | **Average** |
|---|---|---|---|---|---|---|---|---|---|
| 24k | 35.91 | 44.52 | 36.86 | 58.23 | 52.79 | 28.42 | 33.80 | 56.99 | 40.21 |
| 26k | 33.43 | 43.80 | 39.59 | 47.25 | 54.30 | 28.88 | 35.40 | 57.38 | 38.62 |
| 28k | 33.58 | 44.12 | 40.96 | 48.38 | 55.36 | 29.03 | 33.20 | 57.77 | 38.85 |
| 30k | 36.55 | 46.73 | 41.47 | 61.77 | 56.13 | 30.41 | 33.60 | 56.99 | 41.64 |
| 32k | 37.64 | 46.76 | 42.92 | 57.95 | 56.81 | 28.57 | 35.40 | 58.88 | 42.20 |
| 34k | 38.33 | 46.72 | 42.75 | 57.86 | 57.39 | 28.42 | 35.80 | 58.09 | 42.52 |
| 36k | 39.99 | 45.28 | 39.33 | 51.31 | 58.12 | 29.19 | 36.40 | 57.30 | 42.63 |
| 38k | 41.46 | 48.20 | 43.52 | 63.09 | 58.48 | 27.50 | 38.20 | 58.41 | 44.83 |
| 40k | 39.11 | 47.81 | 43.52 | 59.85 | 58.64 | 29.34 | 36.60 | 58.88 | 43.46 |
| 42k | 41.36 | 48.27 | 43.86 | 61.74 | 59.24 | 28.57 | 37.00 | 59.19 | 44.81 |
| 44k | 42.47 | 48.56 | 44.28 | 61.53 | 59.56 | 29.34 | 36.20 | 60.46 | 45.52 |

*Table 35.* Full evaluation results of 6B model width growth with noise std=0.05 in Figure 6

| Steps | MMLU | QA_average | Arc_C | BoolQ | Hellaswag | Logiqa | Openbookqa | Winogrande | **Average** |
|---|---|---|---|---|---|---|---|---|---|
| 24k | 36.03 | 44.81 | 37.03 | 58.47 | 52.72 | 29.34 | 34.00 | 57.30 | 40.42 |
| 26k | 33.94 | 43.63 | 39.08 | 46.94 | 54.30 | 29.95 | 34.20 | 57.30 | 38.78 |
| 28k | 35.60 | 45.12 | 41.72 | 55.57 | 54.94 | 29.03 | 32.80 | 56.67 | 40.36 |
| 30k | 36.87 | 46.48 | 41.72 | 61.35 | 56.09 | 29.65 | 34.20 | 55.88 | 41.68 |
| 32k | 37.89 | 46.45 | 42.75 | 57.37 | 56.84 | 29.49 | 35.00 | 57.22 | 42.17 |
| 34k | 38.30 | 47.89 | 42.41 | 62.54 | 57.00 | 29.03 | 38.60 | 57.77 | 43.10 |
| 36k | 39.65 | 46.35 | 40.27 | 56.57 | 58.26 | 28.42 | 36.00 | 58.56 | 43.00 |
| 38k | 40.02 | 47.57 | 43.00 | 59.85 | 58.62 | 30.41 | 35.00 | 58.56 | 43.80 |
| 40k | 39.72 | 47.93 | 43.43 | 61.10 | 59.00 | 26.88 | 37.60 | 59.59 | 43.83 |
| 42k | 41.85 | 48.21 | 44.28 | 60.34 | 59.16 | 30.11 | 36.80 | 58.56 | 45.03 |
| 44k | 42.41 | 48.42 | 45.31 | 60.06 | 59.39 | 29.65 | 36.20 | 59.91 | 45.42 |

*Table 36.* Full evaluation results of 6B model width growth with noise std=0.1 in Figure 6

| Steps | MMLU | QA_average | Arc_C | BoolQ | Hellaswag | Logiqa | Openbookqa | Winogrande | **Average** |
|---|---|---|---|---|---|---|---|---|---|
| 24k | 35.84 | 44.67 | 36.86 | 58.72 | 52.80 | 28.88 | 34.40 | 56.35 | 40.25 |
| 26k | 34.38 | 43.77 | 38.99 | 46.97 | 54.33 | 29.19 | 34.80 | 58.33 | 39.07 |
| 28k | 35.90 | 44.10 | 41.98 | 47.03 | 55.28 | 29.19 | 33.60 | 57.54 | 40.00 |
| 30k | 37.10 | 46.17 | 41.30 | 60.24 | 56.30 | 29.80 | 33.00 | 56.35 | 41.63 |
| 32k | 38.24 | 47.00 | 43.94 | 61.56 | 56.39 | 28.42 | 34.40 | 57.30 | 42.62 |
| 34k | 38.26 | 47.23 | 43.26 | 61.93 | 57.14 | 28.11 | 36.00 | 56.91 | 42.74 |
| 36k | 40.09 | 45.85 | 42.06 | 54.16 | 58.16 | 26.73 | 36.00 | 58.01 | 42.97 |
| 38k | 41.33 | 46.48 | 43.17 | 54.62 | 58.43 | 29.95 | 35.00 | 57.70 | 43.90 |
| 40k | 39.58 | 48.60 | 44.28 | 62.87 | 59.07 | 28.88 | 37.00 | 59.51 | 44.09 |
| 42k | 41.98 | 48.11 | 43.77 | 61.38 | 59.02 | 28.88 | 36.20 | 59.43 | 45.05 |
| 44k | 41.69 | 48.18 | 44.71 | 57.16 | 59.42 | 29.65 | 37.20 | 60.93 | 44.93 |

*Table 37.* Full evaluation results of 6B models direct growth under different model structure in Figure 16

| Model | Steps | MMLU | QA_average | Arc_C | BoolQ | Hellaswag | Logiqa | Openbookqa | Winogrande | **Average** |
|---|---|---|---|---|---|---|---|---|---|---|
| 3B Pre-norm | 8k | 29.19 | 41.26 | 34.56 | 53.36 | 43.62 | 29.65 | 32.60 | 53.75 | 35.22 |
| | 16k | 32.07 | 44.10 | 37.03 | 56.79 | 50.15 | 29.65 | 34.40 | 56.59 | 38.09 |
| | 24k | 35.83 | 44.42 | 36.69 | 57.95 | 52.90 | 29.19 | 33.80 | 55.96 | 40.12 |
| | 32k | 36.21 | 43.73 | 40.61 | 44.13 | 54.87 | 31.18 | 33.80 | 57.77 | 39.97 |
| 6B Pre-norm Depth | 8k | 29.75 | 41.58 | 34.30 | 59.70 | 43.29 | 29.65 | 29.80 | 52.72 | 35.66 |
| | 16k | 31.23 | 44.10 | 36.35 | 60.73 | 48.80 | 30.72 | 32.80 | 55.17 | 37.66 |
| | 24k | 34.22 | 44.57 | 37.80 | 62.45 | 50.90 | 27.80 | 33.20 | 55.25 | 39.39 |
| | 32k | 34.45 | 43.64 | 40.61 | 49.02 | 53.48 | 28.88 | 33.20 | 56.67 | 39.05 |
| 6B Pre-norm Width | 8k | 30.71 | 41.53 | 34.81 | 54.04 | 43.58 | 30.26 | 32.40 | 54.06 | 36.12 |
| | 16k | 33.23 | 44.04 | 37.03 | 56.79 | 50.13 | 30.26 | 34.00 | 56.04 | 38.64 |
| | 24k | 35.99 | 44.68 | 37.29 | 58.07 | 52.73 | 29.34 | 33.80 | 56.83 | 40.33 |
| | 32k | 35.84 | 43.61 | 40.53 | 44.22 | 54.88 | 30.88 | 33.40 | 57.77 | 39.73 |
| 3B Post-norm | 8k | 28.85 | 40.06 | 29.69 | 59.48 | 41.59 | 26.88 | 30.00 | 52.72 | 34.46 |
| | 16k | 30.17 | 42.27 | 33.02 | 56.33 | 47.01 | 29.95 | 32.60 | 54.70 | 36.22 |
| | 24k | 32.64 | 42.18 | 33.87 | 48.50 | 50.37 | 30.26 | 34.20 | 55.88 | 37.41 |
| | 32k | 31.71 | 42.73 | 36.52 | 45.90 | 52.80 | 29.34 | 35.20 | 56.59 | 37.22 |
| 6B Post-norm Depth | 8k | 27.45 | 37.55 | 30.63 | 55.99 | 36.95 | 22.12 | 28.20 | 51.38 | 32.50 |
| | 16k | 25.17 | 37.23 | 28.84 | 56.57 | 36.25 | 21.66 | 27.80 | 52.25 | 31.20 |
| | 24k | 23.01 | 35.13 | 28.41 | 48.38 | 30.78 | 22.43 | 30.20 | 50.59 | 29.07 |
| | 32k | 23.12 | 35.36 | 27.73 | 51.59 | 31.51 | 21.81 | 29.00 | 50.51 | 29.24 |
| 6B Post-norm Width | 8k | 28.98 | 40.16 | 29.44 | 59.39 | 41.54 | 28.11 | 29.60 | 52.88 | 34.57 |
| | 16k | 30.00 | 42.14 | 33.02 | 56.09 | 46.98 | 29.19 | 32.60 | 54.93 | 36.07 |
| | 24k | 32.51 | 42.13 | 34.04 | 48.81 | 50.44 | 30.57 | 34.00 | 54.93 | 37.32 |
| | 32k | 31.73 | 42.63 | 36.35 | 45.81 | 52.78 | 29.65 | 35.00 | 56.20 | 37.18 |

*Table 38.* Full evaluation results of 17B model pre-training in Figure 1, Figure 12 and Figure 11

| Steps | MMLU | QA_average | Arc_C | BoolQ | Hellaswag | Logiqa | Openbookqa | Winogrande | **Average** |
|-------|------|-----------|-------|-------|-----------|--------|-----------|-----------|---------|
| 0k | 25.00 | 33.33 | 25.00 | 50.00 | 25.00 | 25.00 | 25.00 | 50.00 | 29.17 |
| 4k | 29.90 | 40.95 | 35.67 | 53.76 | 47.10 | 25.96 | 32.60 | 50.59 | 35.42 |
| 8k | 34.23 | 45.25 | 41.64 | 50.21 | 57.81 | 27.04 | 37.20 | 57.62 | 39.74 |
| 12k | 40.15 | 48.97 | 46.08 | 54.71 | 65.56 | 28.26 | 38.20 | 61.01 | 44.56 |
| 16k | 43.33 | 50.77 | 47.87 | 61.99 | 64.98 | 28.11 | 39.40 | 62.27 | 47.05 |
| 20k | 46.40 | 51.72 | 49.49 | 59.85 | 66.57 | 28.88 | 41.20 | 64.33 | 49.06 |
| 24k | 48.58 | 53.57 | 51.62 | 64.10 | 67.69 | 30.72 | 41.80 | 65.51 | 51.08 |
| 28k | 49.17 | 54.10 | 52.13 | 65.87 | 68.20 | 29.80 | 40.80 | 67.80 | 51.64 |
| 32k | 51.11 | 54.35 | 52.22 | 66.30 | 69.46 | 28.11 | 40.80 | 69.22 | 52.73 |
| 36k | 51.71 | 55.02 | 55.80 | 67.43 | 69.91 | 26.73 | 41.20 | 69.06 | 53.37 |
| 40k | 53.64 | 55.55 | 56.14 | 66.18 | 71.66 | 28.88 | 41.40 | 69.06 | 54.60 |
| 44k | 55.70 | 56.64 | 55.29 | 69.24 | 73.08 | 29.80 | 42.20 | 70.24 | 56.17 |
| 48k | 56.02 | 56.90 | 56.91 | 64.53 | 73.78 | 29.65 | 43.20 | 73.32 | 56.46 |
| 52k | 57.61 | 57.98 | 57.51 | 70.70 | 73.35 | 29.49 | 43.80 | 73.01 | 57.79 |
| 56k | 58.03 | 58.53 | 57.68 | 72.29 | 74.15 | 29.49 | 44.00 | 73.56 | 58.28 |
| 60k | 58.57 | 58.53 | 58.87 | 70.34 | 74.56 | 31.49 | 43.20 | 72.69 | 58.55 |
| 62k | 58.75 | 58.55 | 58.70 | 70.00 | 74.54 | 29.80 | 45.40 | 72.85 | 58.65 |
| 64k | 59.17 | 58.10 | 57.25 | 70.34 | 75.04 | 29.34 | 44.40 | 72.22 | 58.63 |
| 66k | 58.16 | 58.47 | 57.42 | 71.04 | 74.25 | 30.72 | 43.60 | 73.80 | 58.32 |
| 68k | 59.65 | 58.51 | 58.45 | 72.45 | 74.84 | 29.80 | 42.60 | 72.93 | 59.08 |
| 70k | 60.06 | 59.12 | 58.02 | 73.12 | 75.00 | 29.80 | 44.40 | 74.35 | 59.59 |
| 72k | 59.64 | 59.21 | 58.70 | 74.04 | 75.11 | 30.26 | 43.40 | 73.72 | 59.42 |
| 74k | 59.42 | 59.53 | 57.76 | 73.58 | 75.31 | 32.10 | 44.00 | 74.43 | 59.48 |
| 76k | 59.71 | 59.38 | 59.22 | 72.91 | 74.99 | 31.18 | 44.40 | 73.56 | 59.54 |
| 78k | 60.75 | 59.56 | 58.28 | 75.99 | 75.12 | 30.26 | 44.00 | 73.72 | 60.16 |
| 80k | 59.93 | 59.07 | 59.13 | 71.47 | 75.91 | 30.57 | 44.20 | 73.16 | 59.50 |
| 81k | 60.24 | 59.35 | 58.19 | 75.60 | 75.12 | 29.80 | 43.80 | 73.56 | 59.79 |
| 82k | 60.49 | 59.24 | 57.85 | 73.24 | 75.45 | 31.03 | 44.40 | 73.48 | 59.87 |
| 83k | 61.61 | 60.50 | 59.64 | 77.71 | 76.17 | 30.41 | 44.00 | 75.06 | 61.05 |
| 84k | 61.74 | 60.20 | 59.39 | 75.02 | 76.11 | 30.26 | 44.80 | 75.61 | 60.97 |
| 85k | 61.74 | 60.45 | 59.81 | 75.41 | 76.37 | 31.03 | 44.80 | 75.30 | 61.10 |
| 86k | 62.47 | 59.55 | 58.02 | 72.94 | 75.97 | 30.88 | 44.80 | 74.66 | 61.01 |
| 87k | 62.05 | 60.23 | 58.45 | 74.71 | 76.45 | 31.80 | 45.00 | 74.98 | 61.14 |
| 88k | 62.14 | 60.68 | 60.15 | 75.41 | 75.59 | 32.10 | 45.40 | 75.45 | 61.41 |
| 89k | 61.76 | 59.69 | 60.41 | 73.79 | 76.29 | 29.19 | 43.80 | 74.66 | 60.73 |
| 90k | 62.29 | 60.53 | 60.07 | 73.94 | 76.12 | 31.49 | 45.80 | 75.77 | 61.41 |
| 91k | 62.09 | 59.75 | 58.79 | 74.25 | 75.94 | 31.49 | 43.60 | 74.43 | 60.92 |
| 92k | 62.10 | 60.06 | 59.30 | 74.80 | 76.13 | 30.57 | 44.60 | 74.98 | 61.08 |
| 93k | 62.32 | 60.25 | 59.64 | 74.86 | 76.50 | 30.26 | 45.60 | 74.66 | 61.29 |
| 94k | 62.14 | 59.89 | 59.39 | 74.31 | 76.16 | 31.03 | 44.40 | 74.03 | 61.01 |
| 95k | 61.94 | 59.90 | 60.15 | 72.29 | 76.66 | 31.49 | 44.80 | 74.03 | 60.92 |
| 96k | 62.93 | 60.14 | 59.04 | 78.59 | 76.53 | 29.95 | 42.60 | 74.11 | 61.53 |
| 97k | 62.70 | 60.13 | 60.32 | 72.42 | 76.65 | 30.72 | 45.60 | 75.06 | 61.41 |
| 98k | 62.95 | 60.06 | 59.81 | 74.50 | 76.35 | 31.80 | 44.80 | 73.09 | 61.50 |
| 99k | 62.86 | 60.34 | 59.81 | 74.83 | 76.15 | 31.03 | 44.20 | 76.01 | 61.60 |
| 100k | 62.77 | 59.87 | 60.84 | 72.29 | 76.08 | 29.34 | 45.00 | 75.69 | 61.32 |
| 101k | 62.63 | 60.11 | 59.90 | 75.11 | 76.85 | 29.03 | 44.80 | 74.98 | 61.37 |
| 102k | 62.40 | 60.21 | 59.98 | 73.73 | 77.00 | 30.57 | 45.00 | 74.98 | 61.31 |
| 103k | 62.45 | 59.96 | 59.98 | 72.72 | 76.60 | 29.65 | 44.40 | 76.40 | 61.20 |
| 104k | 62.63 | 60.78 | 60.41 | 74.16 | 76.64 | 31.95 | 46.40 | 75.14 | 61.71 |

*Table 39.* Full evaluation results of 34B model interpositional growth in Figure 1, Figure 12 and Figure 11

| Steps | MMLU | QA_average | Arc_C | BoolQ | Hellaswag | Logiqa | Openbookqa | Winogrande | **Average** |
|---|---|---|---|---|---|---|---|---|---|
| 60k | 57.96 | 58.05 | 58.10 | 69.02 | 73.93 | 30.72 | 42.60 | 73.93 | 58.01 |
| 61k | 59.61 | 58.64 | 59.04 | 68.47 | 75.86 | 29.95 | 43.40 | 75.14 | 59.13 |
| 62k | 59.54 | 58.60 | 57.85 | 69.48 | 75.46 | 30.11 | 42.80 | 75.88 | 59.07 |
| 63k | 60.25 | 60.27 | 59.47 | 73.24 | 75.84 | 31.80 | 45.60 | 75.69 | 60.26 |
| 64k | 60.51 | 59.66 | 58.45 | 73.15 | 76.18 | 30.88 | 45.60 | 73.72 | 60.09 |
| 65k | 60.57 | 59.29 | 57.85 | 71.22 | 75.92 | 31.18 | 43.80 | 75.77 | 59.93 |
| 66k | 60.19 | 58.90 | 59.30 | 67.80 | 76.06 | 32.41 | 43.40 | 74.43 | 59.55 |
| 67k | 60.54 | 60.04 | 59.40 | 74.25 | 76.18 | 32.41 | 44.20 | 73.80 | 60.29 |
| 68k | 61.50 | 59.40 | 59.47 | 69.82 | 76.19 | 32.72 | 45.20 | 73.01 | 60.45 |
| 69k | 61.42 | 59.53 | 60.67 | 67.49 | 76.65 | 31.95 | 45.60 | 74.82 | 60.48 |
| 70k | 61.67 | 59.66 | 60.49 | 69.11 | 76.74 | 32.26 | 44.60 | 74.74 | 60.66 |
| 71k | 62.08 | 60.36 | 59.98 | 72.57 | 77.09 | 31.64 | 44.40 | 76.48 | 61.22 |
| 72k | 61.97 | 60.34 | 61.01 | 72.05 | 77.03 | 31.95 | 45.00 | 74.98 | 61.15 |
| 73k | 62.21 | 60.29 | 59.90 | 70.80 | 77.06 | 32.26 | 45.00 | 76.72 | 61.25 |
| 74k | 62.74 | 60.99 | 60.67 | 73.15 | 76.97 | 31.18 | 46.40 | 77.58 | 61.87 |
| 75k | 62.85 | 60.67 | 60.15 | 71.41 | 77.33 | 32.10 | 46.00 | 77.03 | 61.76 |
| 76k | 63.13 | 61.19 | 60.49 | 75.60 | 77.42 | 31.03 | 46.80 | 75.77 | 62.16 |
| 77k | 62.92 | 60.41 | 59.81 | 72.05 | 77.09 | 30.57 | 46.80 | 76.16 | 61.67 |
| 78k | 63.52 | 60.05 | 59.90 | 69.88 | 77.47 | 30.11 | 46.20 | 76.72 | 61.78 |
| 79k | 62.65 | 61.27 | 60.84 | 76.18 | 77.11 | 31.49 | 45.20 | 76.80 | 61.96 |

*Table 40.* Full evaluation results of 34B model stack growth in Figure 12

| Steps | MMLU | QA_average | Arc_C | BoolQ | Hellaswag | Logiqa | Openbookqa | Winogrande | **Average** |
|---|---|---|---|---|---|---|---|---|---|
| 60k | 57.86 | 57.55 | 57.17 | 70.24 | 73.99 | 29.34 | 44.40 | 70.17 | 57.71 |
| 61k | 58.41 | 58.45 | 58.36 | 71.74 | 74.84 | 28.73 | 43.20 | 73.80 | 58.43 |
| 62k | 58.52 | 59.24 | 59.56 | 73.85 | 75.14 | 29.95 | 43.60 | 73.32 | 58.88 |
| 63k | 58.63 | 59.49 | 59.81 | 71.41 | 75.22 | 30.41 | 45.20 | 74.90 | 59.06 |
| 64k | 58.79 | 59.47 | 60.04 | 73.31 | 74.66 | 29.98 | 44.20 | 74.65 | 59.13 |

*Table 41.* Full evaluation results of 70B model growth in Figure 1 and Figure 11

| Steps | MMLU | QA_average | Arc_C | BoolQ | Hellaswag | Logiqa | Openbookqa | Winogrande | **Average** |
|---|---|---|---|---|---|---|---|---|---|
| 79k | 62.63 | 61.24 | 61.09 | 76.09 | 77.04 | 32.10 | 44.80 | 76.32 | 61.94 |
| 79.5k | 63.59 | 61.86 | 61.09 | 76.42 | 78.15 | 31.64 | 45.40 | 78.45 | 62.72 |
| 80k | 64.49 | 62.29 | 60.58 | 78.41 | 77.77 | 32.41 | 46.20 | 78.37 | 63.39 |
| 80.5k | 64.44 | 62.07 | 62.03 | 74.95 | 77.89 | 32.10 | 47.20 | 78.22 | 63.25 |
| 81k | 65.08 | 62.20 | 62.17 | 78.23 | 77.74 | 31.64 | 46.80 | 76.64 | 63.64 |
| 81.5k | 64.70 | 62.43 | 62.46 | 77.03 | 78.21 | 31.80 | 46.40 | 78.69 | 63.57 |
| 82k | 64.93 | 62.14 | 62.46 | 76.76 | 78.31 | 32.72 | 44.60 | 77.98 | 63.53 |
| 82.5k | 65.55 | 62.10 | 62.54 | 75.41 | 77.96 | 33.49 | 45.40 | 77.82 | 63.83 |
| 83k | 66.06 | 62.52 | 62.71 | 78.47 | 78.08 | 31.64 | 45.20 | 79.01 | 64.29 |
| 83.5k | 65.85 | 61.62 | 61.69 | 75.29 | 78.01 | 30.41 | 45.40 | 78.93 | 63.74 |
| 84k | 65.87 | 62.48 | 63.05 | 77.86 | 78.38 | 31.64 | 45.40 | 78.53 | 64.17 |

