# OpenReview forum: "Beyond Sunk Costs: Boosting LLM Pre-training Efficiency via Orthogonal Growth of Mixture-of-Experts"
_ICML.cc/2026/Conference — ICML 2026 regular_

### Official Review · Reviewer_R2Ji · 2026-03-08

**Soundness:** 2
**Presentation:** 3
**Significance:** 4
**Originality:** 2
**Overall Recommendation:** 5
**Confidence:** 4

**Summary:**

The paper proposes a strategy called "orthogonal growth" aimed at recycling and leveraging pre-trained Mixture-of-Experts (MoE) model checkpoints by strategically expanding their parameters prior to continued training. It introduces two growth strategies: Depth Growth, which employs interpositional layer copying to increase model depth ; and Width Growth, which duplicates experts and adds minor noise. Although the paper presents a substantial body of experimental results, the core innovation of its primary modules remains rather incremental. Furthermore, the paper lacks in-depth explanations for crucial methodological choices, such as the "orthogonality" of the fused modules and the fundamental reasons why interposition outperforms stacking.

**Compliance With Llm Reviewing Policy:**

Affirmed.

**Final Justification:**

I appreciate the considerable effort you have put into explaining the orthogonality of the fused modules and the reasoning behind why interposition outperforms stacking. Given your willingness to define and describe "orthogonality" with the necessary scientific rigour, I am updating my recommendation to "Accept".

**Key Questions For Authors:**

1. Could you provide a more rigorous, in-depth explanation regarding the orthogonality of fusing these two modules?
2. Could you provide deeper insight or a mathematical justification for why interposition outperforms stacking in the depth expansion module?

**Limitations:**

Yes.

**Strengths And Weaknesses:**

Strengths:
1. The paper tackles the expansion of model depth and width within the MoE framework, which is a highly relevant and important challenge in the field of machine learning.
2. The authors have conducted a vast amount of engineering and experimental work, as evidenced in both the main text and the appendices, with many experiments carried out on a genuinely large scale.

Weaknesses:
1. Methodological contributions for depth expansion and width expansion are incremental.
The proposed "orthogonal growth" is essentially a combination of "depth expansion" (layer copying) and "width expansion" (expert copying). The concept of layer copying for depth growth and duplicating experts and adding noise for width growth has been discussed in previous references. Although not technically the same, from a core algorithmic perspective, these modules do not introduce fundamental innovations.
2. Explanation of module orthogonality is insufficient.
The justification for the orthogonality of the two modules lacks sufficient depth. In Section 3.3, the authors merely state that "depth and width growth are mutually orthogonal" because both the "width-then-depth" and "depth-then-width" paths converge to equivalent performance levels. However, the impact of combining these methods on gradient flow and the optimisation landscape during training is complex. There is a noticeable absence of rigorous analysis to firmly support the conclusion that these strategies are genuinely orthogonal.
3. Explanation for Interposition over Stacking is insufficient.
To argue that interposition is superior to stacking for depth expansion, the authors rely on the empirical observation that the weight norms of layers in pre-trained models show an increasing trend. However, the paper fails to provide a deeper analysis or explanation for why this weight norm gradually increases (and potentially drops slightly at the end). The argument would be significantly stronger with mathematical derivations under specific assumptions or a detailed analysis of the gradient dynamics to justify the choice of interposition.

---

> ### Author Rebuttal · Authors · 2026-03-27
>
> We thank the reviewer for the detailed feedback. We address each concern below:
>
> **| W1: Novelty Justification**
>
> We would like to kindly note that our contribution lies not in any single technique, but in the **systematic study of model growth for well-converged MoE** — a setting prior work has not addressed (existing work focuses on early-stage trained LLMs or dense models). Specifically: (1) the first systematic comparison of depth vs. width growth for converged MoE, (2) the interposition vs. stacking boundary condition (~1×Fc), (3) sunk cost–performance strong positive correlation in growth timing, and (4) large-scale validation (70B, 1T tokens). As Reviewer xQoy noted: *"this paper makes a valuable contribution by systematically clarifying the relationship between these strategies through solid experimentation."*
>
> **| W2 & Q1: Orthogonality Evidence**
>
> By construction, depth and width growth operate on structurally disjoint parameter subspaces (new layers vs. new experts, zero intersection). We agree order-independence alone is insufficient and **have conducted post-hoc gradient and optimization landscape analysis:**
>
> **Evidence I: Gradient direction analysis.** We extracted Adam's first moment (exponential moving average of gradients) from training checkpoints and computed cosine similarity on shared parameters between pre- and post-growth models:
>
> |Steps|cos(m_base,m_depth)|cos(m_base,m_width)|
> |--:|--:|--:|
> |+2k|0.031|−0.003|
> |+4k|0.039|0.022|
> |+6k|0.012|0.034|
> |+8k|−0.001|0.027|
> |+10k|0.012|0.001|
> |+12k|−0.009|−0.018|
> |+14k|−0.008|0.011|
> |+16k|0.016|0.038|
>
> Both growth paths maintain near-zero cosine similarity with the pre-growth gradient direction throughout 16k steps of training (all |cos|<0.04), indicating that each growth operation redirects optimization into a subspace orthogonal to the original trajectory. The orthogonality is stable across all 8 measured checkpoints.
>
> Consistently, the cosine similarity of cumulative weight updates between depth-grown and width-grown models decreases over training (0.63→0.49→0.43), with MoE-specific parameters showing the strongest divergence (expert FFN: 0.61→0.41; router: 0.36→0.32 in gradient space). This indicates that the two growth paths increasingly specialize in different regions of the optimization landscape, particularly on the parameters most central to MoE functionality.
>
> **Evidence II: Growth order does not significantly affect the final model at 12B scale.** We compared the final weights of 12B models grown via depth→width versus width→depth at iteration 54k. The overall cosine similarity is 0.823 with uniform distribution across all 36 layers (range: 0.78–0.86). If the two growth operations were strongly coupled — i.e., one substantially altered the optimization landscape for the other — we would expect the final solutions to differ significantly depending on order. The observed uniformity confirms that the two operations act on largely independent subspaces.
>
> We clarify that "Orthogonal" in our work means *practical orthogonality*: disjoint expansions, near-zero gradient correlation, and decreasing mutual interference. A detailed analysis on this gradient and optimization landscape analysis will be included in the revised version.
>
> **| W3 & Q2: Why Interposition Outperforms Stacking**
>
> As shown in Figure 4, well-trained MoEs universally exhibit a monotonically increasing per-layer weight norm profile. We conducted checkpoint analysis (detailed in our response to Reviewer gRkf Q1, Table R1) showing that **stacking disrupts this smooth profile with a sharp discontinuity at the growth boundary (10× normal inter-layer variation), while interposition preserves it by construction**.
>
> **Theoretical intuition.** Recent work establishes that well-trained residual networks converge to smooth dynamical systems. Marion et al. [1] proved that gradient-flow-trained ResNets are implicitly regularized toward neural ODEs, where transformations $x_{l+1} = x_l + f_l(x_l)$ form a smooth sequence. Wang et al. [2] showed stable deep training requires controlling residual contribution growth across layers.
>
> Thus we hypothesis that the increasing norm profile reflects a fundamental property: deeper layers learn progressively larger perturbations to the residual stream, forming a smooth trajectory. **Stacking disrupts this** by forcing the representation to "jump back" at the boundary,  and the model spends more budget repairing the discontinuity. **Interposition preserves it** by doubling the trajectory's resolution while maintaining continuity.
>
> [1] Marion et al., "Implicit regularization of deep residual networks towards neural ODEs," ICLR 2024.
>
> [2] Wang et al., "DeepNet: Scaling Transformers to 1,000 Layers," IEEE TPAMI 2024.
>
> ---
>
> We hope the comprehensive analyses on spanning gradient flow, optimization landscape, and weight norm continuity successfully address the reviewer's concerns. We remain available for any further discussions or questions.

---

> > ### Author Rebuttal · Reviewer_R2Ji · 2026-04-03
> >
> > Thank you for your detailed and comprehensive response.
> > I appreciate the considerable effort you have put into explaining the orthogonality of the fused modules and the reasoning behind why interposition outperforms stacking. Given your willingness to define and describe "orthogonality" with the necessary scientific rigour, I am updating my recommendation to "Accept".

---

> > > ### Author Response · Authors · 2026-04-04
> > >
> > > Thank you for the thoughtful re-evaluation, and we truly appreciate the time and effort you invested in the detailed review! Your questions on orthogonality and interposition are valuable in pushing us to strengthen the analysis, and we will make sure the revised version incorporates these improvements with full rigor.

---

### Official Review · Reviewer_D78L · 2026-03-13

**Soundness:** 3
**Presentation:** 3
**Significance:** 3
**Originality:** 3
**Overall Recommendation:** 5
**Confidence:** 4

**Summary:**

The paper considers a setting in which practitioners wish to “recycle” existing pre-trained checkpoints for increased efficiency. In particular, there is a smaller LM, trained to near-convergence, and the question is how to “grow” this into a larger LM. The paper considers two directions of growth: increasing the layers, and increasing the experts. For growing layers, they present an interpositional ordering (in contrast to conventional stacking), offering a hypothesis that this retains weight norm patterns. For growing experts, they duplicate experts and add some perturbation to each. Experiments are done with a fixed additional training budget, a fixed total budget, and across model sizes and checkpoints; they demonstrate that growing the smaller model can result in better performance than pre-training the large model from scratch.

**Compliance With Llm Reviewing Policy:**

Affirmed.

**Ethical Review Concerns:**

.

**Final Justification:**

See review.

**Key Questions For Authors:**

* Can you provide more analysis of the width growth method? Namely, why it works and considering other initializations beyond duplication+noise and random.
* For growing from 17 to 70B, is double growth needed? What happens if you grow directly (by setting k=4)?

**Limitations:**

yes

**Strengths And Weaknesses:**

Strengths:
Soundness:
Good validation why depth growth with interposition helps. The paper presents a hypothesis that maintaining a pattern of increasing layer-wise weight norm is important, and that stacking disrupts this while interposition does not. This hypothesis is supported with extensive experiments.
Good characterization of when the proposed approach works. The paper shows that growing the model at different checkpoints results in different rates of improvement.
The paper considers how to combine the two axes of growth.
Empirical validation covers both scenarios of fixed additional and fixed cumulative budget for training.
Presentation:
Paper is well organized and easy to follow. Figures are easy to understand.
Significance:
This paper looks at growing nearly-converged models, which is a realistic setting. For instance, suppose a practitioner has finally found a good training recipe for a small LM and has trained it to completion. The implication of this paper is that rather than pretraining a larger model from scratch using the scaled-up recipe, it might be preferable to simply grow the small LM.
Originality:
The paper focuses on growing nearly-converged smaller LMs, which is different from previous work that considers under-trained LMs. In this setting, existing approaches like stacking layers do not work well, and the paper offers an alternative and a hypothesis for why.

Weaknesses:

Soundness:
For a fair comparison to training from scratch, hyperparameter tuning should be included. I.e., showing that the best training recipe using your technique is better than the best training recipe pre-training from scratch (i.e. not holding batch size, LR, etc. the same across everything).
For width growth, I don’t completely understand the intuition of why the proposed method works. The claim is that exactly duplicating the experts is insufficient, while a small amount of perturbation (0.01-0.1) will improve performance. Why is this the case - do the perturbations allow for fundamentally different experts after training? Things like trying even larger perturbations, or trying other initializations (interpolating the expert params) that are not random but are meant to capture “different but still similar” directions could be interesting.

Soundness*

---

> ### Author Rebuttal · Authors · 2026-03-27
>
> We thank the reviewer for the careful reading and thoughtful questions. We address each point below.
>
> **| Q1: Hyperparameter Tuning**
>
> We deliberately used identical hyperparameters across all methods to ensure a fair ablation of the growth framework itself. This design choice isolates the advantage attributable to initialization (inheriting well-trained weights) rather than learning rate scheduling.
>
> We expect that optimized learning rates would improve both growth and from-scratch baselines, but the growth advantage should persist — as noted in Section 4.2, continuing training from a learning-rate-decayed checkpoint without an appropriate warm-up or reduced learning rate is harmful, and this effect is orthogonal to the growth method.
>
> **| Q2: Other Expert Initialization Methods**
>
> Expert duplication is chosen to maximally inherit optimized parameters from the pre-trained checkpoint. As shown in Figure 6, random initialization performs substantially worse, confirming that knowledge inheritance is the primary driver of growth benefit. Adding a small amount of noise breaks symmetry and encourages expert divergence while preserving the learned representations.
>
> Alternative strategies such as interpolation or orthogonal perturbation are interesting directions to explore. In our setting of expanding a **well-converged** MoE model, **excessive perturbation of trained experts risks significant performance degradation. So we prioritize preserving the well-optimized parameter landscape while introducing just enough asymmetry for expert divergence.** Small noise perturbations do not produce fundamentally different experts at initialization, but instead gradually encourage specialization during continued training，which is a gentler approach well-suited to the converged regime. A systematic comparison of initialization strategies under varying convergence stages would be valuable future work.
>
> **| Q3: Direct 4× Growth**
>
> Direct 4× growth is theoretically feasible. In fact our 70B experiment used sequential 2×→2× (17B→34B→70B) partly for computational reasons, because the intermediate 34B stage itself benefits from sunk cost (150B tokens of additional training). Whether direct 4× matches sequential 2×→2× at this scale is an open question we could not verify due to compute constraints. However, given the orthogonal nature of depth and width growth demonstrated in our analysis, we expect comparable final performance.
>
> ---
>
> We appreciate the constructive feedback and hope the above responses adequately address the concerns. We are happy to discuss further if any questions remain.

---

> > ### Author Rebuttal · Reviewer_D78L · 2026-04-03
> >
> > Thanks!

---

> > > ### Author Response · Authors · 2026-04-04
> > >
> > > Thank you for the feedback, much appreciated!

---

### Official Review · Reviewer_xQoy · 2026-03-13

**Soundness:** 4
**Presentation:** 3
**Significance:** 3
**Originality:** 2
**Overall Recommendation:** 5
**Confidence:** 4

**Summary:**

The authors strive to outline a broad context of computational sustainability in large language model development, addressing the critical challenge of efficiently leveraging pre-trained checkpoints. The article's central area consists of proposing an "orthogonal growth" framework for Mixture-of-Experts (MoE) architectures, introducing two complementary expansion strategies: (1) interpositional layer copying for depth-wise growth, which preserves learned weight-norm patterns in converged models, and (2) noisy expert duplication for width-wise growth, where minimal Gaussian noise facilitates expert specialization without destabilizing pre-trained knowledge. Through systematic scaling experiments up to 70B parameters and 1T tokens, the work demonstrates that recycling sunk computational investment yields a 10.6% accuracy improvement over training from scratch under identical additional compute budgets, while establishing practical guidelines for optimal growth timing relative to training convergence.

**Compliance With Llm Reviewing Policy:**

Affirmed.

**Final Justification:**

My concerns have been addressed.

**Key Questions For Authors:**

- **Alternative Width Expansion Strategies**: For width growth, the current work focuses exclusively on expanding the number of MoE experts. However, other dimensions exist, such as increasing the number of attention heads or directly expanding the hidden dimension. Have you considered how these alternative expansion methods compare to expert duplication in terms of efficiency and performance? Specifically, what is the relationship between expanding MoE width, attention width, and hidden dimensions, and under what conditions might one be preferred over the others? I encourage the authors to discuss this trade-off in future work.

- **Data Distribution Shift**: In practical scenarios, when a well-converged checkpoint is available for growth, the training data distribution may have evolved (e.g., new domains, updated corpora). Therefore, a randomly initialized model trained on the updated data mix might outperform a grown model that inherits biases from the original checkpoint. How do you envision the growth process handling significant data distribution shifts, and does the "sunk cost" correlation hold when the post-growth data differs substantially from the pre-growth data?

**Limitations:**

yes

**Strengths And Weaknesses:**

**Soundness**: The methodology is empirically well-supported with extensive experiments.
- The comparative analysis between interposition and stacking for depth growth is particularly insightful: by correlating layer-wise weight norm distributions with model convergence status and systematically varying the pre-growth FLOPs ratio, the authors clearly delineate the boundary conditions under which interposition outperforms stacking.
- The quantitative boundary analysis (~1× Chinchilla-optimal FLOPs) significantly strengthens the verifiability of the core claim.
- Section 4.2's controlled-computation-budget experiments are practically valuable, directly addressing when practitioners should grow existing checkpoints versus train larger models from scratch.
- The inclusion of ablation studies on growth order (depth-then-width vs. width-then-depth) further solidifies the claims regarding orthogonality.

**Presentation**: The paper is well-organized and clearly written. Figures effectively communicate key insights, particularly the weight-norm visualizations and the growth-timing ablations. The appendix provides extensive evaluation details and hyperparameters, enhancing reproducibility. The distinction between "sunk cost with fixed additional budget" and "fixed total budget" is clearly highlighted.

**Significance**: The problem of computational efficiency in LLM pre-training is undeniably important. The finding that "sunk cost" positively correlates with final performance offers actionable guidance for practitioners managing large-scale training budgets. The focus on well-converged MoE models fills a notable gap, as prior growth work primarily targeted early-stage training or dense architectures.

**Originality**: While model growth and upcycling are established concepts, and specific techniques like layer stacking or expert duplication have been discussed in prior literature, this paper makes a valuable contribution by systematically clarifying the relationship between these strategies through solid experimentation.

---

> ### Author Rebuttal · Authors · 2026-03-27
>
> We sincerely thank the reviewer for the strong endorsement. We address each question below.
>
> **| Q1: Alternative Width Expansion Strategies**
>
> Prior work has explored width expansion through attention heads or FFN hidden dimensions. However, these approaches require padding or projection of existing weight matrices (e.g., Net2Net [1], MSG [2]), which introduces approximation error or alters residual stream statistics. In contrast, expert duplication is a functionally lossless operation that preserves all learned representations. Furthermore, Du et al. [3] show that expanding internal FFN hidden size tends to underperform depth growth in the LLM setting.
>
> Given these considerations, we focus exclusively on expert-count scaling as the width growth axis in this work. We agree that a systematic comparison of multiple width dimensions and their interconnections when expanded jointly would be a valuable direction for future research.
>
> **| Q2: Data Distribution Shift**
>
> This is a practical and important question. Our experiments use the same data distribution to isolate the effect of growth methods. We acknowledge that evaluating behavior under distribution shift is essential for real-world applicability.
>
> **1. When new dataset is of lower quality**. In continued training from a pre-trained checkpoint, the new data may differ in quality or domain. Bae et al. [4] show that lower-quality data can degrade performance during post-growth training (e.g., when practitioners continue training an open-source LLM on curated but noisier corpora).
>
> **2. When new dataset is having a distribution shift**. Wu et al. [5] demonstrate that domain-shifted data of comparable quality can benefit from growth combined with LoRA, acquiring new capabilities while preserving general knowledge. Because the grown model is equipped with additional parameters, new knowledge can be seamlessly integrated through a well-designed training strategy.
>
> We believe our finding of a positive "sunk cost"–performance correlation remains valid under distribution shift, provided data quality is maintained and the training recipe is appropriately adapted. A systematic investigation of growth under diverse data regimes is an important direction for future work.
>
> ---
>
> [1] Chen et al., "Net2Net: Accelerating Learning via Knowledge Transfer," ICLR 2016.
>
> [2] Yao et al. "Masked structural growth for 2x faster language model pre-training." ICLR 2024.
>
> [3] Du et al. "Stacking your transformers: A closer look at model growth for efficient llm pre-training." NeurIPS 2024.
>
> [4] Bae et al. "Relaxed recursive transformers: Effective parameter sharing with layer-wise lora." ICLR 2025.
>
> [5] Wu et al., "LLaMA-Pro: Progressive LLaMA with Block Expansion," ACL 2024.
>
> ---
>
> Once again, we appreciate the constructive feedback! We are willing to discuss further if any questions remain.

---

> > ### Author Rebuttal · Reviewer_xQoy · 2026-04-02
> >
> > Thank the authors for the response, my concerns have been addressed.

---

> > > ### Author Response · Authors · 2026-04-04
> > >
> > > Thank you for the kind acknowledgement. We're glad the response addressed your concerns!

---

### Official Review · Reviewer_gRkf · 2026-03-23

**Soundness:** 3
**Presentation:** 4
**Significance:** 3
**Originality:** 3
**Overall Recommendation:** 5
**Confidence:** 3

**Summary:**

The paper proposes techniques to utilize the "sunk cost" of training smaller models before scaling up to larger models during the development of a large pre-trained LLM. Specifically, these techniques directly use the weights of a smaller model to construct the larger model instead of pre-training from randomly initialized weights. The paper introduces two orthogonal techniques: depth growth (layer copying) and width growth (expert copying). Noticing that the norms of layer weights increase with layer depth in converged models, the authors propose growing depth by "interleaving" layer copies, i.e. duplicating each layer in place, rather than "stacking" them, i.e. concatenating duplicates of the entire model end-to-end. The authors also propose growing width by duplicating experts (to increase the total number of experts) with a small amount of noise. The paper shows that utilizing these techniques to first train on smaller models leads to better final model quality for both a fixed additional training budget and a fixed total training budget.

**Compliance With Llm Reviewing Policy:**

Affirmed.

**Final Justification:**

Thanks to the authors for the thoughtful responses! All my concerns have been addressed and I maintain my original recommendation to accept.

**Key Questions For Authors:**

1. Could you explain the reasoning behind the ~3e20 additional training budget for the fixed additional budget experiments?
2. How do you think a model with the same number of parameters, but increased layer width, would compare to the grown model architecture (when both are trained from scratch)?
3. How much more do you think width growth and depth growth could be applied to further grow models (trained to a similar model quality of a model at the grown size trained from scratch), while still saving computational costs? Do you think these techniques can be effectively used to grow models beyond 4x their original size?
4. Just curious - do you think something like muP could help generalize the learning rates / learning rate schedules?

**Limitations:**

yes

**Strengths And Weaknesses:**

This paper proposes a simple but novel and effective method to save LLM pre-training costs. Overall, it is well-written and shows strong experimental results.

Soundness
- Overall, the paper's claims are well-supported, particularly by extensive experimental results.
- Though an interesting observation, I'm not sure if the weight norms increasing by layer depth necessarily indicate that interposition would be more effective than stacking. The authors claim that this pattern doesn't emerge/stabilize until later in training, but the plot for 0.6*Fc already appears to have the increasing weight norm pattern. As the layernorms enforce scale invariance, would it mainly be the magnitudes themselves affecting the training of the grown models? I'm curious as to what the weight norms look like throughout the training of both the interpositional and stacked grown models.
- I wonder if the additional training budget for the fixed additional training budget experiments is sufficient for the 6B model, as it doesn't appear to be the same order of magnitude of the largest sunk cost models. It would be interesting to see how the grown models would compare with a significantly larger training budget.
- The authors honestly note that they use a constant learning rate for grown models, which may not be the most effective value or schedule. I wonder if the grown models could show even better performance with muP (maximal update parameterization).
- The pre-trained from scratch baseline at the grown model size uses the same architecture as the grown models. I wonder how a model with the same number of parameters, but with increased layer width would compare to this baseline. (As I understand, this would be a more typical way of scaling up the model.)

Presentation
- The paper's structure and writing are easy to follow. It provides sufficient detail for reproduction of results, and enough background for the reader to contextualize its results. (However, I will admit that I am not particularly familiar with the area.)
- Nit: there are a few minor typos that can easily be fixed.

Significance
- The paper provides a readily-usable simple method to save computational resources for scaling up LLMs by utilizing the "sunk cost" of previously-trained smaller models. This is generally applicable to large scale LLM pre-training.
- However, it appears to be difficult to use this method to indefinitely scale up LLMs, as it does not allow for increasing layer dimensions. (The paper experiments with up to 4x the original model size.) Nevertheless, this could be an interesting future direction.

Originality
- The paper demonstrates that directly reusing weights from smaller models can be an effective way of training larger LLMs, and introduces and justifies novel techniques for doing so.

---

> ### Author Rebuttal · Authors · 2026-03-27
>
> We thank the reviewer for the positive assessment and insightful questions.
>
> **| Q1: Do magnitudes affect training of grown models?**
>
> We hypothesize that norm magnitude is an important factor beyond the trend shape alone. At 0.6×Fc, the increasing weight norm pattern has already emerged but with small magnitude; consequently, stacking's disruption of this trend can be recovered relatively quickly as norms continue to grow during continued training.
>
> As you suggested, we compare weight norm patterns across base 3B, stacking 6B, and interposition 6B:
>
> > Table R1: Per-layer average expert weight norm at selected layers
>
> |Layer|Base 3B|Stack@Growth|Stack+16k|Inter@Growth|Inter+14k|
> |--:|--:|--:|--:|--:|--:|
> |0|.0323|.0323|.0352|.0323|.0349|
> |10|.0336|.0336|.0363|.0337|.0363|
> |**19**|**.0356**|**.0356**|**.0373**|.0336|.0362|
> |**20**|—|**.0323↓**|**.0367↓**|.0336|.0362|
> |30|—|.0336|.0373|.0347|.0370|
> |39|—|.0356|.0373|.0356|.0376|
>
> Stacking introduces a sharp norm discontinuity at the growth boundary (L19→20: drop of 0.0033, **10× the average inter-layer difference**). After 16k training steps, the gap diminishes but persists (0.0006). Interposition preserves the smooth monotonic profile with no boundary artifact (max jump = 0.0014, identical to base). This magnitude mismatch accounts for stacking's inferior performance: the model must expend training budget repairing the discontinuity, and larger norm magnitudes amplify this repair cost.
>
> **| Q2: Reasoning behind the ~3e20 additional training budget**
>
> For 3B we pre-trained for ~4e20 FLOPs and allocated ~4e20 FLOPs for post-growth training, simulating a half-and-half scenario. As growth timing moves later, sunk cost increases to 8e20–1.2e21, making the additional budget proportionally smaller. But actually this is consistent with our Section 4.2 recommendation: **post-growth budget should be at least on the same order as sunk cost.** When budget ≫ sunk cost, the grown model serves as better initialization; when budget ≪ sunk cost, continued training is insufficient to realize the growth benefit.
>
> **| Q3: Performance with muP**
>
> This is a valuable open research question. muP could potentially benefit model growth by maintaining stable learning dynamics across scales. Our initialization strategies (layer duplication, expert copying) are agnostic to parameterization and should be compatible with muP in principle. However, since our setting focuses on converged models, additional learning rate tuning may be required for already-optimized parameters, which is not addressed by muP's pre-training focus. We consider this an important direction for future work.
>
> **| Q4: More methods for width expansion**
>
> Expert duplication is a functionally lossless operation that preserves all learned representations. In contrast, expanding hidden dimensions requires padding or projection of existing weight matrices (e.g., Net2Net [1], MSG [2]), which introduces approximation error or alters residual stream statistics. Du et al. [3] further show that expanding internal FFN hidden size tends to underperform depth growth in the LLM setting. We agree that a systematic comparison would be valuable, but since prior work has extensively studied attention and dense FFN dimension expansion, we focus on expert scaling for MoE models.
>
> **| Q5: Can we indefinitely scale up? Beyond 4×?**
>
> Exploration beyond 4× growth is valuable, though practical limits exist:
>
> 1. **Depth scaling limits.** Extremely deep networks face trainability challenges: gradient pathologies become harder to control even with LayerNorm, and signal propagation degrades with depth [4]. Current LLM architectures rarely exceed ~128 layers, suggesting a practical ceiling for pure depth growth.
>
> 2. **Width (expert count) scaling limits.** Adding experts increases memory and communication overhead; load balancing also becomes harder as expert count grows [5]. Since the hidden dimension remains fixed, per-expert capacity is bounded — beyond a certain point, adding more small experts yields diminishing returns.
>
> We believe 8–16× total growth is feasible with careful scheduling (alternating depth and width stages), but the optimal recipe remains an exciting open direction.
>
> **| Q6: Minor typos**
>
> Thank you for the careful reading. We will conduct a thorough review and correct all typographical errors in the revised version.
>
> ---
>
> [1] Chen et al., "Net2Net: Accelerating Learning via Knowledge Transfer," ICLR 2016.
>
> [2] Yao et al. "Masked structural growth for 2x faster language model pre-training." ICLR 2024.
>
> [3] Du et al. "Stacking your transformers: A closer look at model growth for efficient llm pre-training." NeurIPS 2024.
>
> [4] De & Smith, "Batch Normalization Biases Residual Blocks Towards the Identity Function in Deep Networks," NeurIPS 2020.
>
> [5] Fedus et al., "Switch Transformers: Scaling to Trillion Parameter Models with Simple and Efficient Sparsity," JMLR 2022.
>
> ---
>
> Once again, we appreciate your constructive feedback!

---

> > ### Author Rebuttal · Reviewer_gRkf · 2026-04-03
> >
> > Thank you for your response! My concerns have mostly been addressed, but I just wanted to clarify one of my questions as I think I didn't phrase it clearly.
> >
> > In my second question (fourth point under soundness, which I believe the authors addressed as Q4), I did not mean to ask about other methods of width expansion that use previously trained weights, but about whether the architecture of the grown model makes the best use of additional parameters from the perspective of training from scratch. I think it would be interesting to include an "optimal" or "standard" model architecture (trained from scratch) for a given parameter count as an additional baseline, though I can understand that it may be difficult to determine / out of the scope for this paper. (For instance, in the Chinchilla model family, the 6.7B model has both more layers and a larger hidden dim than the 3.5B model.)

---

> > > ### Author Response · Authors · 2026-04-04
> > >
> > > Thank you for the clarification, we appreciate the thoughtful follow-up.
> > >
> > > Indeed, our grown model inherit the hidden dimension of the base model, which may not match the "optimal" architecture for the target parameter count. When it comes to how to best allocate additional parameters, we believe that simultaneously expanding both dimensions is a natural and sensible strategy because of the orthogonal nature of depth and width growth demonstrated in our work. In fact, if the base model already has a well-balanced depth-to-width ratio, a simultaneous double growth would preserve this ratio, which we believe is a practical and reasonable default. This is also aligned with our motivation of doule growth strategy as we explored in our scalability experiments. Finer-grained control over the allocation (e.g., which specific layers to replicate and which experts to duplicate) involves a larger design space that is beyond the scope of this paper.
> > >
> > > More broadly, how to optimally distribute additional parameters across depth and width dimensions remains a valuable open question, and we hope our framework provides a useful foundation for future exploration in this direction. In conclusion, we sincerely appreciate the reviewer's constructive engagement throughout this discussion!

---

### Decision · Program_Chairs · 2026-04-30

**Decision:**

Accept (regular)

**Comment:**

The paper proposes a new growth strategy that uses checkpoints of smaller pre-trained models to build larger MoE models. The issues raised were largely addressed during the rebuttal, and all reviewers unanimously voted to accept the paper.